

**A non-stationary model for reconstruction of historical annual**
**runoff on tropical catchments under increasing urbanization**
**(Yaoundé, Cameroon)**
Camille Jourdan[1,2], Valérie Borrell-Estupina[3], David Sebag[4], Jean-Jacques Braun[5,6], Jean-Pierre Bedimo
Bedimo[6,†], François Colin[2], Armand Crabit[2], Alain Fezeu[7], Cécile Llovel[8], Jules Rémy Ndam Ngoupayou[9],
Benjamin Ngounou Ngatcha[10], Sandra Van-Exter[11], Eric Servat[1], Roger Moussa[2]
[1] OSU OREME, Univ Montpellier, Montpellier, France
[2] LISAH, Univ Montpellier, INRA, IRD, SupAgro, Montpellier, France
[3] HSM, Univ Montpellier, CNRS, IRD, Montpellier, France
[4] Normandie Univ, UNIROUEN, UNICAEN, CNRS, M2C, Rouen, France
[5] GET, CNRS, IRD, University of Toulouse, Toulouse, France
[6] Institut de Recherches Géologiques et Minières, Centre de Recherches Hydrologiques, Yaoundé, Cameroon
[7] French National Research Institute for Development (IRD), Yaoundé, Cameroon
[8] WSP France, Toulouse, France
[9] Laboratoire de Géologie de l'ingénieur et d'Altérologie, Département des Sciences de la Terre et de l'Univers,
Faculté des Sciences, Université de Yaoundé I, BP 812, Yaoundé, Cameroun
[10] Department of Earth Sciences, Faculty of Sciences, University of Ngaoundéré, Ngaoundere, Cameroon
[11] GM, Univ Montpellier, CNRS, Université des Antilles, Montpellier, France
† deacesed
*Correspondence:* Camille Jourdan (camille.jourdan@umontpellier.fr)





**Abstract.** Inter-tropical regions are nowadays faced to major land-use changes in data-sparse context leading to
difficulties to assess hydrological signatures and their evolution. This work is part of the theme *Panta Rhei* of the
IAHS, and aims to develop a combined approach of data acquisition and a new semi-distributed model taking into
account land-use changes to reconstruct and predict annual runoff on an urban catchment. Applications were
conducted on the Mefou catchment at Nsimalen (421 km²; Yaoundé, Cameroon) under rapid increase in
urbanization since 1960. The data acquisition step combines an historical data processing and a short-term
spatially-dense dedicated instrumentation (2017-2018), leading to 12 donor catchments, 6 from historical studies
and 6 from the instrumentation presenting various topographic, soil and land-use characteristics. We developed an
annual rainfall-runoff model based on mathematical relationships similar to the SCS model. The model needs the
definition of a hydrological index $I$ which is time variable and enables to take into account land-use changes and
non-stationary relationships between rainfall and runoff. The index $I$ is an empirical indicator defined as a
combination of several components such as topography, soil, and land-use. The rules for the construction of $I$ are
obtained from data analysis on donor catchments. Then, the model was calibrated on donor catchments. Finally,
two applications were conducted on eight target catchments composing the Mefou in order: i) to study the spatial
hydrological functioning and calculate the water balance during the short instrumentation period; ii) to reconstruct
the hydrograph at the Mefou and to simulate the impact of future scenarios of land-use and urbanization. Results
show that that the Mfoundi catchment, integrating the three more urbanized sub-catchments, contributes near to
40 % of the Mefou despite covering only 23 % of the basin. The most urbanized sub-catchments present annual
runoff coefficient about 0.86 against 0.24 for the most natural sub-catchments. The second result is the
reconstruction of historical annual runoff from 1930-2017 with $r^2 = 0.68$, $RMSE = 99$ mm and a mean absolute
normalized error $\bar{E} = 14.5$ % over the 29 observed years. The reconstruction of the annual runoff at Nsimalen
confirms the moderate impact of urbanization on annual runoff before 1980. However, a decrease of about 50 %
of the forest cover and an increase from 10 % to 35 % of the urban area between 1980 and 2017 are associated
with an increase of 53 % of annual runoff coefficient for the Mefou at Nsimalen (0.44 against 0.29). Application
for a fictive plausible scenario of urbanization in 2030 leads to an increase of more than 85 % of the annual runoff
in comparison of the values observed in 1980. The coupled experimental-modelling approach proposed herein
opens promising perspectives regarding the evaluation of the annual runoff in catchments under changes.




## 1    Introduction


The link between the hydrological cycle and human societies has been strong with changes and
intensification of these interactions over time (Koutsoyiannis, 2013; McMillan et al., 2016). In response to the
imperative to include human increasing impacts as integral to hydrological research, the International Association
of Hydrological Sciences (*IAHS*) launched the hydrological decade (2013-2022) with theme "*Panta Rhei: Change*
*in Hydrology and Society*". Due to rapid and complex anthropic changes, the *IAHS* emphasize the necessity to
improve the capability of decision maker and water resources stakeholders to make predictions of hydrological
dynamics and support sustainable societal development in a changing environment (Montanari et al., 2013).
Quantifying and understanding past changes in hydrological processes are necessary to suggest reliable future
predictions of hydrological signatures. Reconstructing past data and predicting annual, monthly and daily
hydrographs in a changing environment, and especially on poorly gauged catchments with sparse data, remains a
challenge for hydrological science.
Long-term hydrological modelling requires integrating the impact of global changes in terms of climate,
land-use and infrastructures. Nowadays, urban areas represent only 2 % of the total surface of the Earth but
concentrate more than 50 % of world population, cities count close to four billions people, this figure was
multiplied by five since 1950 (Janicot et al., 2015). This huge urbanization rate combined with a demographic
explosion is especially significant in the inter-tropical regions where most of developing countries and numerous
in development megalopolis are located (UNDESA, 2017). For example, population in Africa is projected to reach
2.5 billion people by 2050 with about 55 % living in urban areas  (Güneralp et al., 2017). Hence in this context,
the impact of land-use changes on runoff, especially in urban and peri-urban zones, seems to override rainfall
changes impacts.
Empirical, conceptual, probabilistic and physically-based models can be used to simulate the impact on
runoff of global changes. Conceptual models such as *HBV* (Bergström and Singh, 1995), *GR1A*, *GR2M* or *GR5J*
(Mouelhi, 2003; Mouelhi et al., 2006; Le Moine, 2008) or physically-based models as *MIKE-SHE*
(Abbott et al., 1986) were applied to assess global changes impacts on hydrology. On tropical climate, such models
were applied at the local scale (Giertz et al., 2006), the mesoscale (Beck et al., 2013; Wagner et al., 2013;
Yira et al., 2016) or large scale catchments (Genwei, 1999; Zhou et al., 2010) at daily, monthly or annual time
steps. All these models require accurate information about the basin physiographic characteristics, long series of
rainfall-runoff data, climate and land-use changes data; moreover an adequate calibration/validation strategy must
be undertaken in order to take into account the spatio-temporal evolution of some parameters. However, on basins
with sparse data at various time steps (e.g. only monthly rainfall available on a given period), and in the absence
of continuous long series rainfall-runoff data, simple modelling approaches must be adapted for reconstructing
annual runoff taking into account available sparse historical data and information on climate and land-use changes.
For that, empirical approaches in non-dimensional spaces were largely used since Turc (1954) and Budyko (1974)
which were largely applied, analysed and extended the last decade (Zhou et al., 2015; see a synthesis
in Moussa and Lhomme, 2016). Ponce and Shetty (1995) developed an original annual rainfall-runoff model based
on a formulation similar to the one developed in the Soil Conservation Service method (SCS, 1956;
Mishra and Singh, 2013), and Sivapalan et al. (2011) extended this approach to model different components of the
water cycle at the annual scale. However, all these models generally don't take into account changes in climate



and land-use. Hence, there is a need to develop simple parsimonious approaches modelling for annual runoff taking
into account non-stationarity due to land-use long term evolution, and adapted to basins with sparse data.
In most of developing countries, environmental monitoring as precipitation and streamflow are often limited.
This data-sparse condition results of a poor knowledge of basin climatology and hydrological signatures, such as
annual runoff. Even for areas faced with recurring high water management issues, most of national organizations
do not have resources to purchase and maintain the necessary instrumentation for field monitoring
(Hughes et al., 2015). The availability of continuous and long term data sets of runoff varies dramatically
throughout the world (Kundzewicz et al., 2007). Prediction in ungauged basins (*PUB*) approaches are tools to cope
with this data-sparse context (Blöschl, 2013) and are based on regionalization of hydrological characteristics by
spatial proximity or geomorphological similarities from donor to target catchments (Parajka et al., 2013; Salinas
et al., 2013). However, recently development of soft monitoring (Crabit et al., 2011) and crowdsourced hydrology
(Lowry and Fienen, 2013; Le Coz et al., 2016; Mazzoleni et al., 2017) gave encouraging results. To cope with the
lack of long-term observation (rainfall-runoff) on catchment faced to land-use changes a solution is to set up a
dedicated short-term instrumentation on catchment faced to various land-use states associated with a best
valorisation of historical database. This methodology lets to observe hydrological processes of catchments
characterized by various states of land-use and under various climatic contexts.
The scope of this paper is to develop a combined approach of data acquisition and the development of a new
semi-distributed model taking into account land-use changes to reconstruct and predict annual runoff on a
catchment exposed to high urban increase. The data acquisition step implies (1) to deploy a complementary and
dedicated short-term and multi-scale space hydro-meteorological network, (2) to analyse the most recent global
land-use products with adapted time and space resolution and (3) to maximize the valorisation of historical studies
for the evaluation of catchment characteristics (land-use, topography, soils map) and some environment variables
(evaporation, precipitation, runoff). The model developed from the data acquisition is mainly based on land-use
changes impacts on annual runoff.
The Mefou catchment (421 km²) including the city of Yaoundé capital of Cameroon was used for applications
to reconstruct the annual runoff at the outlet for the period 1930-2017. A dedicated hydrological instrumentation
during the hydrological year 2017-2018 was conducted, and completes the review of historical studies over the
period 1960-2016.
First, we introduce the annual non-stationary rainfall-runoff model structure, hypotheses and general
calibration-validation procedure. The inputs of the model are the annual precipitation and an hydrologic index
taking into account topography, soil and land-use temporal evolution. Second, we describe the study area including
the review of historical hydro-meteorological data and a description of the dedicated short-terms instrumentation.
Third, we present a spatio-temporal analysis of precipitation. Fourth, we present the methodology to construct the
hydrological index from the analysis of hydrologic and physiographic data. Then, we present the model
parametrization, calibration and validation, and finally we show and discuss the application results of
reconstruction and prediction of historical annual runoff. The Appendix A gives the list of notations and
abbreviations. The Supplementary Material gives additional information on data sets.





## 2    The annual rainfall-runoff model

### 2.1    Model structure

The model is based on mathematical relationship between precipitation $P$ and runoff $R$ similar that proposed by Ponce and Shetty (1995) and Sivapalan et al. (2011) for applications at the annual scale on the basis of the *SCS* equations (Mishra and Singh, 2003). $P$ ranges between $P_n$ and $P_x$ which respectively correspond to the minimal and the maximal precipitation values over a large historical period of the main study catchment. Applications were conducted in tropical basins where $R$ is not nil. Therefore, in order to simplify, we use a simple second order polynomial relationship between $R$ and $P$ such as (Fig.1):

$$R = AP^2 + BP,\qquad(1)$$

and the annual runoff coefficient:

$$\rho = R/P = AP + B,\qquad(2)$$

where $A$ and $B$ are empirical parameters which can be linked to catchment properties (e.g. topography, soil, land-use).

The first hypothesis is that the annual volume $V_O$ at the outlet of the main catchment is the sum of the annual volumes $V_i$ on each sub-catchment $T_i$ (Fig. 2b):

$$V_O = \sum_{i=1}^{n} V_i ,\qquad(3)$$

where $n$ is the number of sub-catchments and $i$ is the index representing a sub-catchment noted $T_i$. We define the annual runoff $R_i$ on each sub-catchment $T_i$ as:

$$R_i = V_i / A_i ,\qquad(4)$$

where $A_i$ is the area of $T_i$ and $A_O$ is the area of the whole catchment with:

$$A_O = \sum_{i=1}^{n} A_i ,\qquad(5)$$

Consequently the annual runoff $R_O$ of the whole catchment is defined as:

$$R_O = V_O / A_0 .\qquad(6)$$

The second hypothesis presumes that the runoff coefficient $\rho = R / P$ and the annual runoff $R$ are only functions of $P$ and an "hydrological index" noted $I$ with $\rho = f(P, I)$ and $R = g(P, I)$ similar to the SCS approach used in Ponce and Shetty (1995) and Sivapalan et al. (2011). As for the curve numbers $CN$ in the Soil Conservation Service method (Mishra and Singh, 2003), the index $I$ characterizes topography, soil and land-use of the basin, and enables to take into account land-use evolution through time: $I$ is considered low for permeable soils and/or low urbanization areas producing low runoff, and $I$ is considered high for impermeable soils and/or high urbanization areas producing high runoff. The index $I$ is an empirical indicator and can be defined as a linear combination of several components $C_i$. As for the $CN$ method, in this study we choose the following three components: topographic component ($C_T$) such as slope classes impacting runoff, and soil component ($C_S$) such as permeability classes and land-use component ($C_{LC}$) such as urbanization classes.





$I = \sum_{i=1}^{m} \omega_i C_i$ ,                                                                     (7)
where $\omega_i$ is the weight attributed to component $i$ and $m$ the number of components including in the hydrological
index. If $m$ is equal to 1, the hydrological index $I$ is based on only one component, for example the land-use
characteristics ($C_{LC}$) if this descriptor is considered as the main factor changing in time.
We note $I_n$ and $I_x$ respectively the lowest and the highest values of $I$ over the catchment dataset used to
construct the model. Therefore, for a given rainfall $P$, $R$ increases when $I$ increases; for a given index $I$, $R$ increases
when $P$ increases.
For a given value of $I$ the runoff coefficient $\rho$ increases when the rainfall increases from $P_n$ to $P_x$. Let $\rho_{n,I}$ and
$\rho_{x,I}$ be the corresponding values of $\rho$ for $P_n$ and $P_x$ respectively. Let $\beta_1 = \rho_{x,I} - \rho_{n,I}$ as shown in Fig. 1a. For a given
value of $P$, the runoff coefficient $\rho$ increases when $I$ increases from $I_n$ to $I_x$. Let $\rho_{n,P}$ and $\rho_{x,P}$ be the corresponding
values of $\rho$ for $I_n$ and $I_x$ respectively. Let $\beta_2 = \rho_{x,P} - \rho_{n,P}$ as shown in Fig. 1a. As we have a linear relationship
between $P$ and $\rho$ (Fig. 1a), the value of $\beta_2$ is constant and similar for all values of $P$. In order to calculate $\beta_2$, we
need data on different catchments and periods with the same value of $P$ but with different values of $I$ ranging
between $I_n$ and $I_x$.
The annual runoff model proposed herein uses a simple relationship $R = f(P, I)$ as shown in Fig. 1b. The
domain of application of the model is for the precipitation $P \in [P_n, P_x]$ and the hydrological index $I \in [I_n, I_x]$. The
model needs as input the precipitation $P$ which has to be calculated on each sub-catchment $T_i$ (see an application
in Sect. 4). The model needs also the definition of the hydrological index $I$ which is time variable and enables to
take into account land-use changes and non-stationary relationships between $R$ and $P$. The definition of the rules
to construct the components $C_i$ and the weights $\omega_i$ of $I$ (Eq. 7) are obtained from data analysis on the study site as
shown later in Sect. 5.

**2.2    Calibration, validation, reconstruction procedure**
The annual rainfall-runoff model developed herein is calibrated using data from donor catchments (noted $D$)
historical information, and from the dedicated short-term instrumentation. Donors could be catchments or sub-
catchments inside or near to the main study catchment (Fig. 2a). Sets of annual precipitation $P$ and annual runoff
$R$ data are available for donors characterized by different geomorphologic and land-use states (e.g. topography,
soil, urbanization). Selecting distant past, recent past and present in $P$ and $R$ values from donors enable to cover a
large range of climate, land-use and geomorphological conditions in order to elaborate a reliable and robust model.
Donors are used to calibrate the model parameters then the calibrated model is applied on target catchments (noted
$T$, Fig. 2b) and let to evaluate runoff for the main catchment (Eq. 3 to 6). The model can be applied for several
periods (past, present and future) for different climate and land-use scenarios.
We consider the set of sparse annual rainfall-runoff data. The set is split into two datasets, the first one is
used for the calibration of the model parameters and the second one for the validation. The dataset used for the
calibration must include data from different ranges of precipitation $P_n < P < P_x$ and land-use characterized by the





hydrological index $I_n < I < I_x$. The *RMSE*, the $r^2$, the normalised error ($E$) and the mean normalised absolute error
($\bar{E}$) criteria functions are used to assess and compare simulation performance:
$$RMSE = \sqrt{\frac{1}{n}\sum\left(\widehat{R_{O\,j}} - R_{O\,j}\right)^2}\,, \qquad (8)$$
$$r^2 = \frac{\left[\sum\left(\widehat{R_{O\,j}} - \overline{\widehat{R_O}}\right)\left(R_{O\,j} - \overline{R_O}\right)\right]^2}{\sum\left(R_{O\,j} - \overline{R_O}\right)^2}\,, \qquad (9)$$
$$E_j = \frac{\widehat{R_{O\,j}} - R_{O\,j}}{R_{O\,j}}\,, \text{ and } E_{i,j} = \frac{\hat{R}_{i,j} - R_{i,j}}{R_{i,j}}\,, \qquad (10)$$
$$\bar{E} = \frac{1}{n}\sum\left(\left|E_j\right|\right)\,, \qquad (11)$$
where $\widehat{R_{O\,j}}$ is the simulated annual runoff and $R_{O\,j}$ the observed annual runoff for the main study catchment; *n* the
number of the evaluated year and *j* the index corresponding to a given year. $\hat{R}_{i,j}$ is the simulated annual runoff for
target *i* for the evaluated year *j* and $R_{i,j}$ the observed annual for target *i* for the evaluated year *j*. In order to evaluate
the robustness of the model, a sensitivity analysis is conducted on the impact of the number of donor catchments
used establishing the rules of the hydrologic index *I*. Finally, the performance of the developed model is compared
to a classical annual runoff model generally applied under stationary conditions (i.e. the *GR1A* model based on the
Turc (1954) equation; Mouelhi, 2003).


**3    Study site**
**3.1    Oro-hydrography and climate**
The Mefou River is a tributary of the Nyong. The Mefou catchment at Nsimalen (421 km²) includes the
capital city of Cameroon, Yaoundé (Fig. 3a).The upstream part of the basin (70 km²) is controlled by the Mopfou
dam built in 1969 planned to provide about one third of the drinking water to the Yaoundé urban area
(100,000 m³.day⁻¹).
The catchment is hilly (peaks at 1000 m a.s.l) with important wetland areas (around 700 m a.s.l) at the
downstream parts (Fig. 3a). The Mefou River is 35 km length from the Mopfou dam to Nsimalen. The main
tributaries of the Mefou is the Mfoundi which drains the most urbanized parts of the whole catchment (Fig. 3a).
The river channel slope ranges between 1 ‰ and 5 ‰ causing frequent floods in the lowlands. Canalization of the
upstream Mfoundi and its tributaries were undertaken since 2002 in order to reduce floods in the urbanized zone.
The landform of the South Cameroon Plateau corresponds to the dismantling of an old iron crust undergoing more
humid climatic conditions (Bilong et al., 1992; Beauvais, 1999; Bitom et al., 2004). This multi-convex landform
is composed of rather closely spaced hilly compartments, typically of few hundred metres in diameter, separated
by flat swampy valleys of variable stretch from 50 to 500 m width (Bitom et al., 2004). We used the slope index
$S_I$ of Roche (Roche, 1963) to characterize the topography component of the hydrological index *I* for donor and
target catchments.
The climate is humid tropical with two dry and wet seasons (wet-and-dry equatorial savannah with dry winter
according to the Köppen-Geiger classification, Kottek et al., 2006). The mean annual precipitation $P_m$ from Mvan




station ($P_1$) in Yaoundé on the period 1930-2015 is 1580 mm. We distinguish four seasons: long rainy season from
March to June, short dry season from July to August, short rainy season from September to November, and long
dry season from December to February. The hydrological year is defined from March to February. Ikounga (1978)
has estimated the potential evapotranspiration *PET* between 900 mm (Sunken Colorado pan) to 1200 mm
(Thornthwaite method). Supplementary Material (Sect. 2, Fig. S1) shows the mean monthly precipitation,
temperature and *PET*.

### 3.2    Soil and land-use

The regolith is developed on a granito-gneissic basement. Ferralsol (laterite) regolith is developed on the
hillslopes while in the swampy valleys, it is topped with bleached hydromorphic soils developed on colluvium and
river alluvium (Bachelier, 1959; Braun et al., 2005; Braun et al., 2012). In the region of Yaoundé,
Humbel and Pellier (1969) calculated a soil surface permeability between 20 and 70 cm.h$^{-1}$ up the hill, and 200
cm.h$^{-1}$ near the swampy valleys. These values of permeability are very high and limit the surface runoff, especially
in swampy valleys. The clay amount is generally higher at the top of the hills than at the bottom. In the field
experiment we conducted in 2017, we measured in the region of Yaoundé the soil surface permeability by a
simplified Beerkan method (Bagarello et al., 2014) and obtained values ranging between 2 and 125 cm.h$^{-1}$ which
are comparable to the values given by Humbel and Pellier (1969). Humbel and Pellier (1969) also showed that for
both types of soil, the surface permeability decreases quickly with the depth until an impermeable layer facilitating
lateral flow. In this study we use the proportion of hydromorphic soil (*HS*) to characterize the soil component of
the hydrological index *I* for donor and target catchments (see Sect. 5.1).
The administrative urban area of Urban Community of Yaoundé (*CUY*) covers nowadays about 297 km². As
most part of the Nyong basin (Olivry, 1979), the Mefou catchment was originally mainly covered by humid tropical
forest. The study area is faced to major land-use changes due to human activities mainly urbanization and
agriculture (see more details in Supplementary Material, Sect. 3). Population in Yaoundé has increased from
90,000 in 1960 (Franqueville, 1968) to 3.65 million in 2017 (UNDESA, 2017) with an annual growth rate of 5.7 %
per year between 1987 and 2005 according to the Central Office of Cameroonian Population Study and Census.
This huge demographic change is characterized by an important expansion of the urban area Fig. 3b and the
increase of population density (Bopda, 2003). In the opposite, forest and wetlands areas decreased, and were
generally replaced by agricultural and urban areas as shown in Fig. 3c with the land-use classification over the
Mefou catchment from the land cover product of the European Space Agency available for Africa (*ESA-CCI LC*)
for the year 2016. Moffo (2017) analysed a set of aerial photography in 1956 and estimated that the impervious
areas covered 3.5 km², less than 1 % of the Mefou catchment area. We used *ESA-CCI LC* and OpenStreetMap®
2015 layers to calculate impervious areas of around 64 km² (15 %) in 2016. Ebodé (2017) used Landsat images to
study the evolution of land-use from 1978 to 2015. He noticed at the Mefou catchment until the Nyong confluence
(basin area of 802 km², approximately two times the Mefou basin area at Nsimalen) a decrease of 160 km² of the
total forest cover, with specifically a decrease of 60 % of the primary forest area from 235 km² in 1978 to 94 km²
in 2015, a decrease of 73 % for swampy forest from 206 km² to 57 km², and an increase of 60 % in degraded and
secondary forest from 223 km² to 353 km². For the Mefou at Nsimalen, Ebodé (2017) estimated that the agricultural



area increased from 10 km² (3.5 % of the catchment area) in 1978 to 28 km² (10 %) in 2015, and that the urban
area (integrating impervious surfaces) increased from 45 km² (11 %) in 1978 to 151 km² (36 %) in 2015. We used
the proportion of urban area $U$ over donor and target catchments to characterize the land-use component of the
hydrological index.

The impact of these land-use changes on hydrological processes is not yet quantified on the Mefou

catchment. The urbanization of the Mefou catchment also impacts both groundwater and river water quality due
to domestic and industrial untreated wastewater from urban areas but also contamination by peri-urban agriculture.
For example, Branchet et al. (2018) recently shows high Diuron® concentration on surface water that frequently
exceeded the European water quality guideline. These growing issues of water management drive the removal of
wetlands in lowlands impacting their ecosystem services as the natural purification of water (Daily, 1997; Russo,

2013).


### 3.3    Historical sparse data

Precipitation measurements are available at a monthly time step at two historical raingauges (Fig. 3a): $P_1$

from 1930 to 2017, and $P_2$ from 1955 to 1978. The correlation coefficient for the common period between both
stations at the annual time step is 0.74. A long-term reference precipitation dataset was calculated using the mean
of $P_1$ and $P_2$ when data from both stations are available, and from $P_1$ for the remaining periods.

Daily runoff measurements at the Mefou catchment outlet at Nsimalen started in 1963 but with long periods

of gaps. Annual runoff is available for only 29 years (1964-1977, 1979, 1982-1986, 2005-2011 and 2017), and
ranges between 250 mm and 850 mm. Annual runoff coefficient ranges between 0.21 and 0.48.

Few studies are available on the hydrology of Yaoundé, and most of them date before the 80[th]. They

particularly focused on water balance at monthly and annual scales: on the Mefou river (Lefèvre, 1966; SNEC,
1969; Olivry, 1979), on the Mfoundi (Srang, 1972; Nguemou, 2008), and on sites downstream Nsimalen
(Ikounga, 1978). From these studies, we retain six historical donors (noted $D_H$; Fig. 4a), and Table 1 presents their
characteristics: area between 24 and 235 km², period of observation, annual precipitation $P$ between 1640 and
1930 mm, annual runoff $R$ between 392 and 1340 mm, and annual runoff coefficient $\rho$ between 0.22 and 0.77.
Under the hypothesis that the storage annual variation is nil, we estimate an annual evapotranspiration $AET = P - R$
which ranges between 400 and 1400 mm. Information relative to precipitation measurements are well documented
in the historical studies previously cited. Note that the donor $D_{H6}$ is located out of the Mefou catchment but quite
close from Yaoundé area (40 km) and presents a similar topography, soil and land-use conditions of the Mefou
catchment (Ikounga, 1978). These donors cover different land-use states: e.g. forestry natural cover for $D_{H4}$ and
$D_{H6}$ ($U < 1\%$) and highly urbanized cover for $D_{H3}$ ($U > 75\%$).

Other historical studies of smaller ($< 10$km²) and larger ($> 5000$ km²) catchments in Nyong basins in natural

land-use context give  $P$ between 1420 and 1730 mm, $R$ between 392 and 530 mm, $\rho$ between 0.18 and 0.30 and
$AET$ between 1070 and 1470 mm. The urbanized catchment Odza (6 km²), located in Yaoundé (Mfoundi





catchment), was monitored in 2011-2012 by Ngoumdoun (2013) who calculated $P = 1840$ mm, $R = 1640$ mm,
$\rho = 0.88$ and $AET = 220$ mm (see Supplementary Material, Sect. 4).

### 3.4 Dedicated short-term multi-scale instrumentation (03/2017-02/2018)

In order to complete historical data, we undertook dense spatial rainfall-runoff instrumentation during one
hydrological year (03/2017-02/2018). Eleven daily time step raingauges were installed in order to study the spatial
variability of precipitation (Fig. 4a). The choice of limnimetric stations location was determined by the position of
the main confluences, by the position of historical limnimetric stations, and the need to measure runoff from basins
with different degrees of urbanization (Fig. 4a). This instrumentation provides six additional experimental donors
(noted $D_I$) with different ranges of heterogeneities in terms of area, land-use, topography and soil (Table 1). The
limnimetric station $D_{I1}$ is located downstream the dam and enables to measure the outflow from the reservoir at
$100 \pm 25$ mm.yr$^{-1}$. The lack of measurements of the outflow value until 2017-2018 makes this results the first
assessment of the dam impact on Mefou water budget. The donor $D_{I4}$ corresponds to the intermediate basin between
$D_{I3}$ and $D_{I2}$. Table 1 presents their characteristics: area between 21 and 120 km², $P$ between 1620 and 1715 mm, $R$
between 712 and 1250 mm, $\rho$ between 0.40 and 0.76 and $AET$ between 405 and 908 mm. Annual precipitation on
these donors $D_I$ are of the same order as for historical donors $D_H$ while runoff and runoff coefficients are generally
higher for catchments with higher urbanization rate.

### 3.5 Main characteristics of the donor catchments

The Mefou catchment originally covers by dense tropical forest includes the most part of the city of Yaoundé.
The urban area started growing since 1960 (1 % of the total basin area) to currently reach about 30-35 % of the
basin area with an impervious area estimated to 15 %. The forest cover has vanished of more than 50 % since 1980
with a huge conversion of the primary forest into secondary and degraded forests. Nowadays, the forests cover
about 40 % of the Mefou at Nsimalen. The various sources cited in Sect. 3.2 showed a growth of agricultural areas
around the urban area of Yaoundé, with cropland and grassland covers around 30 % of the catchment area in 2015.
The catchment can be considered as peri-urban due to the noticeable urbanization and the development of
agricultural activities observed in lowlands and outskirts. However, the south-west part and the area drained by
the Mopfou dam in upstream remain slightly affected by urbanization.
Combining historical studies and dedicated short-term instrumentation 2017-2018, we have 12 donor
catchments, 6 from historical studies ($D_H$) and 6 from the instrumentation ($D_I$) presenting various topographic, soil
and land-use characteristics; the area ranges between 21 and 235 km², $P$ ranges between 1620 to 1930 mm, $R$
between 390 and 1340 mm, $\rho$ between 0.22 and 0.77 and $AET$ between 400 mm to 1400 mm. $S_I$ varies from 6.7 %
to 13.5 %, $HS$ varies from 0 to 44 % and $U$ varies from 0 to 83 % over donor catchments of the study area. We
observe that $\rho$ can vary widely for the same catchment function of the land-use: e.g. from 0.33-0.4 for $D_{H1}$ ($U =$
5 %) and $D_{H2}$ to 0.77 for $D_{H3}$ ($U > 75$ %). All the observed data on the study site are analysed to understand





hydrological processes of the catchments faced to land-use changes in order to identify rules defining the
hydrological index *I* on donors and apply it on targets for the period 1930-2017.


## 4    Precipitation

The model needs as input the precipitation *P* which has to be calculated on both donor and target catchments
(Fig. 4). For donors, precipitation information are well documented in corresponding studies or issue of dedicated
short-term instrumentation (Table 1). For targets, we used the long-term historical raingauges and the spatially
short-term information (1968-1969 from SNEC (1969), and 2017-2018 from the short-term instrumentation) to
construct historical precipitation database for each target.

### 4.1    Temporal variability of precipitation

First, we study long-term precipitation trends (1930-2015) over the Mefou catchment from historical
raingauges $P_1$ and $P_2$. The average precipitation is 1580 mm ($P_m$) and the minimal and maximal annual
precipitation are respectively $P_n$ = 1050 mm and $P_x$ = 2200 mm. Values of $P_n$ and $P_x$ set the limit of availability of
the developed model for the study catchment. Fig. 5a shows no significant trends of annual precipitation over the
period, but we observe a succession of humid (1960-1970, 1980-1990, and 2006-2013) and dry (1935-1950, 1970-
1980, and 1990-2000) periods. At the seasonal scale we observe some changes in amount of precipitation: i) no
change during the first wet season (March to June) (Fig. 5b); ii) during the first dry season (July and August),
increase of the mean precipitation from 100 to 220 mm (+120 %) on the period 1930 to 2015 (Fig. 5c); iii) during
the second wet season (September to November) slight increase from 700 to 760 mm (+9 % ; Fig. 5d); iv) during
the second dry season (December to February) decrease from 110 to 80 mm (-28 %; Fig. 5e). This historical
precipitation database is used to construct the database precipitation for target catchments. Results shows also that
there is no a clear changes on annual precipitation between 1930 and 2015, and consequently the trend of annual
runoff coefficient increase can be related mainly to catchment change and particularly to the increase of urbanized
areas.

### 4.2    Spatial variability of precipitation

Second, we study the spatial distribution of annual precipitation over the Mefou catchment.
Figures 6a and 6b show the mean annual precipitation for respectively the hydrologic years 1968-1969
(SNEC, 1969) and 2017-2018 (dedicated short-term instrumentation), the only historical years with available
dense spatially measured precipitation.
For 1968-1969, *P* varies over the Mefou catchment between 1400 mm to 2000 mm with an average of 1780
mm. For 2017-2018, *P* varies between 1400 and 2100 mm with an average of 1640 mm. The hydrological year



2017-2018 seems quite representative of an average year in terms of annual precipitation ($P_m$ = 1580 mm). The
annual precipitation observed at $P_1$ is about 1730 mm for the same period with a difference of +6 % with average
precipitation $P$. The rain-gauge $P_1$ seems quite representative of the average value for the Mefou catchment.
On both Figs. 6a and 6b, we observe that the highest annual precipitation values are located in the north-west
part of the basin, corresponding to the zone with highest elevations. The east and south parts, corresponding to the
flattest and lowest elevation parts of the basin are characterized by lower $P$.
Due to the lack of spatial information for the historical period, a precipitation weight $w_{Ti}$ is assigned to a
target catchment $T_i$ such as:
$\qquad P_{Ti} = w_{Ti}.P$ ,                                                                                      (12)
where $P$ is the mean annual precipitation on the Mefou catchment from the historical database and $P_{Ti}$ the mean
annual precipitation on $T_i$. The term $w_{Ti} = P_{Ti}/P$ can be both calculated using historical data (Figs. 6a and 6c) and
the instrumentation 2017-2018 (Figs. 6b and 6d). For both cases, we obtain comparable and retain $w_{T1} = w_{T3} = w_{T5}$
$= w_{T6} = w_{T7} = 1$ for respectively $T_1$, $T_3$, $T_4$, $T_5$, $T_6$ and $T_7$; $w_{T2} = 1.05$ for $T_2$ is slightly high due to the high values
of $P$; $w_{T4} = w_{T8} = 0.95$ for $T_4$ and $T_8$.

### 4.3    Relationship between annual runoff coefficient and precipitation

Third we study the relationship between the annual runoff coefficient $\rho$ and $P$ for three stations in nearly-
steady land-use states (Fig. 7): the Mefou at Nsimalen on the period 1964-1984 with a low impact of land-use
evolution in comparison to the period 1984-2015, the Mefou at Etoa natural forested basin on the period 1967-
1983, and the semi-urbanized Mfoundi on the period 1969-1971. Both the Mefou at Etoa and Mfoundi are sub-
catchments of the Mefou at Nsimalen. For the Mefou at Nsimalen, we adjusted a linear relationship between
$\rho$ and $P$; $\rho$ increases from $\rho_{n,P} = 0.2$ corresponding to $P_n = 1050$ mm to $\rho_{x,P} = 0.35$ corresponding to $P_x = 200$ mm;
this gives $\beta_I = \rho_{x,P} - \rho_{n,P} = 0.15$. For the Mefou at Etoa, $\rho = 0.15$ for $P = 1330$ mm and $\rho = 0.30$ for $P = 2200$ mm;
both values of $\rho$ for the Mefou at Etoa are inferior than those observed on the Mefou at Nsimalen for similar values
of precipitation, showing the impact of land-use with low annual runoff coefficient on natural basins. For the
Mfoundi, $\rho = 0.33$ for $P = 1640$ mm and $\rho = 0.40$ for $P = 1930$ mm; both values of $\rho$ for the Mfoundi are superior
than those observed on the Mefou at Nsimalen for similar values of precipitation, showing the impact of land-use
with high annual runoff coefficient on semi-urbanized basins. In summary, we obtain approximately similar values
of $\beta_I = 0.15$ on the Mefou at Nsimalen (1964-1984), but also on the natural Etoa catchment (1967-1983) and on
the in-urbanization Mfoundi upstream catchment (1969-1970). These three catchments have three different values
of the hydrological index $I$, but with $I$ considered constant on the presented periods.




## 5    The hydrological index I

This section analyses the hydrological and physiographic data in order to define the rules for constructing the hydrological index $I$ calculated as a linear combination of three components:

$$I = \omega_T C_T + \omega_S C_S + \omega_{LC} C_{LC} \ , \tag{13}$$

where $C_T$ is the topography component, $C_S$ the soil component, $C_{LC}$ the land-use component, $\omega_T$ the weight of $C_T$, $\omega_S$ the weight of $C_S$, and $\omega_{LC}$ the weight of $C_{LC}$.

### 5.1    The components $C_T$, $C_S$ and $C_{LC}$

Heterogeneities of topography, soil condition and land-use variability in space and time observed on the study area (Sect. 3) lead us to propose classification rules to highlight the main features of catchments and to define the three components $C_T$, $C_S$, and $C_{LC}$ of the hydrological index $I$ using cartographical data. All three components range between 0 and 1; when any of the terms ($C_T$, $C_S$ and $C_{LC}$) increase, $\rho$ increases. The topography ($C_T$) and soil condition ($C_S$) are considered stable over the time contrary to land-use ($C_{LC}$) faced to major changes.

For topography $C_T$, the slope index $S_I$ of Roche (Roche, 1963) is calculated for donors (Table 1) and targets from SRTM (2014). We define: $C_T = 0$ for $S_I < 7$ %; $C_T = 0.5$ for $7\ \% \leq S_I \leq 12\ \%$; $C_T = 1$ for $S_I > 12$ %.

For soil condition $C_S$, the lack of accurate soil maps over the catchment constrains us to define indirectly soil condition heterogeneities over the catchment. Historical studies of soil characteristics (Bachelier, 1959; Pellier, 1969; Humbel and Pellier, 1969) are used to define soil classes depending on the altitude and the slope derived from the SRTM 2014. For that, we calculate the area of lowlands (altitude < 730 m) with low slopes (< 7 %) corresponding to hydromorphic yellow soil characterized by lower rate in clay and higher surface permeability. The proportion of hydromorphic soil ($HS$) on each catchment is used to estimate the classes of $C_S$ (see for donors Table 1, and for targets: $C_S = 0$ for $HS > 15$ %; $C_S = 0.5$ for $2\ \% \leq HS \leq 15\ \%$; $C_S = 1$ for $HS < 2$ %.

For land-use component $C_{LC}$, historical references and global products (summarized in Table 2) are used to characterize land-use of donors and the evolution of target land-use over past-period with available data (1930, 1950, 1980, 2000, and 2017) and future scenario (2030). In order to integrate the main land-use signature, we define six classes for $C_{LC}$ according to urban area proportion of ($U$; see for donors Table 1, and for targets the Supplementary Material, Table S3): $C_{LC} = 0$ for $U < 1$ %; $C_{LC} = 0.2$ for $1 < U < 5$ %; $C_{LC} = 0.4$ for $5 < U < 20$ %; $C_{LC} = 0.6$ for $20 < U < 50$ %; $C_{LC} = 0.8$ for $50 < U < 70$ %; $C_{LC} = 1$ for $U > 70$ %. Different trends of land-use changes are observed for the 8 target sub-catchments (Fig. 8). From 1930 to 1950, the whole main catchment is considered to be mostly cover by originally forest ($C_{LC} = 0$ for all the targets). Development of urbanization impacted first the Mfoundi sub-catchments from 1960 to 1980 ($T_5$, $T_6$ and $T_7$) especially $T_5$. Nowadays, these sub-catchments reach a maximum of urbanization for $T_5$ and $T_6$ ($C_{LC} = 1$). $T_2$ and $T_8$ faced to major changes since 1980 with intensification since 2000. Nowadays, the urbanization process do not get the entire area of these catchments. The urbanization continue and will be amplified in these surrounding areas due to the lack of space in the most urbanized part of Yaoundé ($T_5$, $T_6$ and $T_7$). $T_1$, $T_3$ and $T_4$ are the last sub-catchments impacted by urbanization; a high proportion of these catchments have forest or wetland covers. We propose a fictive but plausible scenario of land-use for 2030, regarding to the current expansion of the urban area and the perspective of future population





estimation for Yaoundé (from 3.6 million in 2017 to 6.7 million in 2035; UNDESA, 2017). We suppose a high
development of urbanization for $T_2$ and $T_8$ ($C_{LC}$ from 0.6 in 2017 to 0.8 in 2030) and in a lesser extent a development
of urbanization over the south part of the basin ($T_4$ with $C_{LC}$ from 0 in 2017 to 0.4 in 2030). Values of $C_T$, $C_S$ and
$C_{LC}$ are presented for donors in Table 3 and for targets in Table 4.

### 5.2 Relationship between annual runoff $\rho$ and $C_T$, $C_S$ and $C_{LC}$ for donors

$D_{H4}$, $D_{H5}$ and $D_{H6}$ correspond to forested areas ($C_{LC} = 0$) whereas $D_{I5}$ and $D_{H3}$ have high rates of urbanization
($C_{LC} = 1$, Table 3). For these basins, precipitation presents a low range of variation between 1645 and 1810 mm.
$D_{H4}$, $D_{H5}$ and $D_{H6}$ present low values of annual runoff coefficient with $\rho$ varying from 0.22 to 0.25. $D_{I5}$ and $D_{H3}$
present very high values of runoff with $\rho = 0.76$ and 0.77. For catchments with intermediate levels of urbanization
($D_{I2}$, $D_{I3}$, $D_{I4}$, $D_{I6}$, $D_{H1}$, $D_{H2}$), runoff ranges between those observed in the two previous cases with $\rho$ ranging
between 0.33 and 0.54.
Analysis of $\rho$ for donors $D_{I2}$ and $D_{I6}$ presenting the same value of $C_{LC}$ (0.6) but extreme values of $C_T$ and $C_S$
($D_{I2}$ is located in the hilly part of the Mefou catchment whereas $D_{I6}$ present high portion of lowlands) enables to
quantify the impact of $C_S$ and $C_T$ on $\rho$. For the period September-December, $D_{I6}$ presents runoff coefficient of 0.40
which is significantly lower than the value of 0.53 observed for $D_{I2}$ on the same period. Differences observed are
quite clear in term of runoff with for $D_{I4}$ runoff value up to 160 mm in October against 95 mm for $D_{I6}$ (see
Supplementary Material, Sect. 5).
These results show the significant impact of land-use conditions on runoff, but topography and soil condition
could explain complex hydrological responses. Consequently when calculating the index $I$, we will give a higher
weight to the component $C_{LC}$ in comparison to $C_T$ and $C_S$.

### 5.3 The weights $\omega_T$, $\omega_S$ and $\omega_{LC}$

From data analysis (Sect. 5.2), we showed that the impact of land-use change on runoff is higher than the
impact of soil and topography. Consequently, we affect higher weight for $\omega_{LC}$, with $\omega_{LC} > \omega_T$ and $\omega_{LC} > \omega_S$.
Figure 9 shows an example of the relationship between annual runoff of donors $\rho$ and hydrological index for
donors $I$ for $\omega_{LC} = 7/9$, $\omega_T = 1/9$ and $\omega_S = 1/9$. We observe a simple linear empirical relationship to estimate $\rho_D$ :
$$\rho_D = aI + b \tag{14}$$
with $r^2 = 0.96$. We conduct a sensitivity analysis of the regression on the adjusted parameters $a$ and $b$ for four sets
of parameters $\omega_{LC}$, $\omega_T$ and $\omega_S$ : i) $\omega_T = 1/5$, $\omega_S = 1/5$ and $\omega_{LC} = 3/5$; ii) $\omega_T = 1/7$ $\omega_S = 1/7$ and $\omega_{LC} = 5/7$; iii), $\omega_T =$
1/9, $\omega_S = 1/9$ and $\omega_{LC} = 7/9$; iv) $\omega_T = 1/12$, $\omega_S = 1/12$ $\omega_{LC} = 10/12$. We obtain very good correlation coefficient of
the four linear regressions ranging between 0.83 (case i) and 0.98 (case iv). The parameter $a$ ranges between 0.66
and 0.83, and the parameter $b$ between 0.10 and 0.19 (note that for $I = 0$ we have $\rho = b$). For $I = 1$, we have $\rho = a$





$+ b$ which varies between 0.80 and 0.84. Around a mean value of $I = 0.5$, all four configurations give $\rho = 0.51$ (See
Supplementary Material, Sect.6).

For the configurations ii, iii and iv, we have $r^2 = 0.96$ +/- 0.02. In the following, we retain the intermediate

set $\omega_T = 1/9$, $\omega_S = 1/9$ and $\omega_{LC} = 7/9$ (corresponding regression presented in Fig.9). Table 3 presents the values of
$I$ for donors and Fig. 10 gives the temporal evolution of $I$ from 1930 until now for targets. Note that in 2017, the
values of $I$ are particularly high for the target catchments $T_5$, $T_6$ and $T_7$ on the Mfoundi basin due to high
urbanization. In contrast, some target catchments such as $T_1$ and $T_4$ are or not impacted nowadays by urbanization
and presents very low values of $I$. Finally, the target catchments $T_2$, $T_3$, and $T_8$ are currently faced to the most
important land-use change and have intermediate values of $I$.

### 5.4    Introducing $I$ in the model structure

In the following, we choose a simple linear relationship between $I$ and $\rho$ (Eq. 14; Fig. 9) which leads that the

value of $\beta_1$ is constant and similar for all values of $I$.

We observe that the impact of land-use change (represented by $I$) on annual runoff ($\beta_2$) is higher than the

impact of precipitation change ($\beta_1$), with $\beta_1 = 0.15$ (Fig. 7), $\beta_2 = 0.60$ (Fig. 9) and $\beta_1 << \beta_2$. We consider a
reference precipitation $P_R = \frac{P_x + P_n}{2}$, and let $\rho_D$ be the runoff coefficient calculated by the linear regression adjusted
from donors under precipitation near of $P_R$ (1625 mm). Then $\rho$ is calculated as the sum of $\rho_D$ and a factor $G$ taking
into account the impact of precipitation:
$$\rho = \rho_D + G ,\tag{15}$$
with
$$G = \frac{\beta_1}{(P_x - P_n)}\left[ P - \frac{P_x + P_n}{2} \right] .\tag{16}$$

For a given value of $I$ (Fig. 9): for $P = P_n$, we have $G = -\beta_1/2$ and consequently $= \rho_D - \frac{\beta_1}{2} = \rho_{n,P}$ ; for

$P = P_x$, we have $G = \beta_1/2$ and consequently $\rho = \rho_D + \frac{\beta_1}{2} = \rho_{x,P}$.

Introducing Eq. (14) and (16) into Eq. (15), we obtain very simple second order polynomial relationship

between $R$ and $P$ (Eq. 17 similar to Eq. 1) and a linear relationship between $\rho$ and $P$ (Eq. 18 similar to Eq. 2), and
$$R = AP^2 + BP ,\tag{17}$$
$$\rho = AP + B ,\tag{18}$$
with $A = \frac{\beta_1}{P_x - P_n}$    and    $B = aI + b - \frac{\beta_1(P_x + P_n)}{2(P_x - P_n)}$ .    (19)

In summary, the model needs the precipitation $P$ as input and the three parameters $a$, $b$ and $\beta_1$ characterizing

the relationship between $\rho$ and $I$. These three parameters can be calibrated using data from donor catchments.





## 6 Applications

The model presented in Sect. 2 is function of precipitation $P$ and the hydrological index $I$. Precipitation was calculated on the target sub-catchments using historical precipitation dataset and the relationships established in Sect. 4. The hydrological index $I$ is defined in Sect. 5 and presented in Fig. 10 for target catchments for the period 1930 - 2030.

First a sensitivity analysis was conducted and the calibrated parameters $a$, $b$ of the model are discussed (Sect. 6.1). Then two applications were conducted on the Mefou at Nsimalen subdivided into eight target sub-catchments (Fig. 4.b) in order:

- to study the spatial hydrological functioning of the basin on eight target sub-catchments and calculate the water balance during the short instrumentation period 2017-2018 (Sect. 6.2).
- to reconstruct the hydrograph at the Mefou at Nsimalen and on the eight sub-catchments for the historical period 1930-2017 and to simulate the impact of future scenarios of land use and urbanization (Sect. 6.3).

### 6.1 Sensitivity analysis, calibration, validation and model comparison

Applications were conducted on the period 1930-2017, using precipitation data on $P_1$ and $P_2$, to reconstruct annual runoff for all eight target sub-catchments and the whole Mefou catchment at Nsimalen. Predictions for the impact assessment of future land-use scenario on annual runoff were then also made. In the application, we distinguish two cases, before and after the dam construction (1970). Before 1970, the catchment $T_1$ (controlled area of the dam location) is considered as other catchments ($R$ depends of $I$ and $P$). After 1970, the simulated $R$ of $T_1$ corresponds to the proportion of precipitation discharged measured during the short-term instrumented period ($\rho = 0.05$ to $0.15$; see Table 1 for $D_{II}$ and Sect. 3.4).

From data analysis in Sect. 4.3 (Fig. 7), we retain $\beta_I = 0.15$. We run a sensitivity analysis on the remaining two parameters $a$ and $b$ (adjusted from the regression $\rho_D = aI + b$ with $I$ calculated using Eq.13 with $\omega_T = 1/9$ and $\omega_S = 1/9$, $\omega_{LC} = 7/9$) for different sets of donor catchments. We run the model for $n = 6, 8, 9$ and 10 donors (see Table 1 and Sect. 3.3 and 3.4). In each run, we select randomly 30 sets of $n$ donors, and in order to have a wide range of variation of $I$, we add a constrain that for at least one point we have $I < 0.3$ and for at least one point we have $I > 0.7$. The model output is given by Eq. 1 to 6 at the Mefou at Nsimalen, and the model is evaluated using the three criteria $RMSE$ (Eq. 8) and $r^2$ (Eq. 9) and $\bar{E}$ (Eq. 11) for the 29 observed years (see Sect. 3.3). For $n = 6$, 8, 9 or the 10 donors, we observe a low impact of the number of donors on the calibrated parameters ($a$ and $b$) and the three performance criteria with $a = 0.68 +/- 0.02$, $b = 0.12 +/- 0.01$, $RMSE = 101 +/- 1$ mm, $r^2 = 0.66$, and $\bar{E} = 15$ % (see Supplementary Material, Sect. 6). The low variability of the average of parameters $a$ and $b$ from $n = 8$ lead us to select 8 donors by keeping the last two donors for validation.

In order to get a model adapted to various states of urbanization, the calibration and validation dataset at the Mefou catchment scale should include periods of low and high urbanization rate. Observed annual runoff at the Mefou at Nsimalen are used in alternate years for calibration (15 years) and validation (14 years). From the sensitivity analysis, we calibrate $a$ and $b$, choosing the set of 8 from 10 donors giving the lowest values of $RMSE$




on the 15 years calibrated period. We use the 9 donors $D_{H2}$, $D_{H3}$, $D_{H4}$, $D_{H6}$, $D_{I2}$, $D_{I3}$, $D_{I4}$, $D_{I5}$ and $D_{I6}$. We obtain
$a = 0.74$; $b = 0.12$ with performance criteria $RMSE = 70$ mm, $r^2 = 0.79$ and $\bar{E} = 11$ %. Figure 9 presents the linear
regression for the calibrated parameters and Fig. 11 shows the abacus $\rho = g(P,I)$ for these parameters. In the
abacus, we plotted the donor catchments by specifying the corresponding estimation of $I$ in parenthesis to compare
with the model. We also plotted the points of Etoa for the period 1967-1983 characterized by a stationary value
of $I$ (0.11) but with a wide range of $P$ (1320 to 2150 mm).

The validation is made at two levels. First, the two remaining donor catchments $D_{H3}$ and $D_{I2}$ are used to

validate; we obtain $E_{D_{H3}} = +8$ % (+105 mm) and $E_{D_{I3}} = +12$ % (+110 mm). Second, at the Mefou at Nsimalen for
the remaining 14 years; we obtain $RMSE = 123$ mm, $r^2 = 0.60$, and $\bar{E} = 18$ %.

The semi-distributed model results were also compared to the stationary lumped annual runoff model $GR1A$

(Mouelhi, 2003) using the same calibration and validation procedure. $GR1A$ is used to compare with a stationary
approach of the catchment characteristics. Results are shown on Fig. 12 with performance for $GR1A$ significantly
lower with $RMSE = 126$ mm, $r^2 = 0.43$ and $\bar{E} = 19$ % for calibration and $RMSE = 128$ mm, $r^2 = 0.42$ and $\bar{E} = 22$ %
for validation over the both same periods at Mefou catchment scale. As the $GR1A$ was calibrated using alternate
years on the whole period, we observe that $GR1A$ slightly overestimate the runoff for the period 1963-1980 (with
low impact of urbanization), and underestimates runoff for the period 2011-2017 (with high impacts of
urbanization)

### 6.2    Annual water balance on the instrumented period 2017-2018

The rainfall-runoff data from the short-term instrumentation 2017-2018 enables to measure the contribution

of the catchments $T_1$, $T_2$, $T_3$ and $T_6$ of the Mefou catchment at Nsimalen. However, the target sub-catchments $T_4$,
$T_5$, $T_7$ and $T_8$ were not gauged. Table 5 gives the values of $P$, $R$, $AET$, $\rho$ and the contribution of each sub-catchment
$K_i$, corresponding to runoff volume of sub-catchment $V_i$ divided by the volume at Nsimalen $V_O$ (with $K_i = V_i / V_O$).
$P$ ranges between 1580 and 1715 mm, $R$ between 100 and 1325 mm, $AET$ between 320 and 1260 mm, $\rho$ between
0.21 and 0.76 and $K$ between 2.5 to 18.5 %. We can classify the eight target catchments into four categories
according to land-use: i) controlled by the dam, $T_1$; ii) urbanized $T_5$, $T_6$ and $T_7$; iii) peri-urban $T_2$, $T_3$ and $T_8$; iv)
natural basins $T_4$.

The first category concerns the sub-catchment $T_1$ controlled by the dam. The annual discharge $R_1$ measured

at the outlet of the dam is 100 mm +/-25 mm, corresponding to a contribution $K_1 = 2.5$ % on the total volume at
Nsimalen.

The second category corresponds to sub-catchments firstly faced to urbanization during the study period. We

have respectively $R_5 = 1230$ +/- 125 mm, $R_6 = 1130$ +/-125 mm and $R_7 = 1030$ +/- 200 mm. The contribution of
$T_5$, $T_6$ and $T_7$ on the total volume at Nsimalen are respectively $K_5 = 18.0$ %, $K_6 = 10.0$ % and $K_7 = 11.0$ %. These
three catchments are characterized of by very low $AET$ between 390 and 620 mm and very high $\rho$ between 0.62
and 0.76.



The third category corresponds to peri-urban catchments. Runoff on $T_2$ (corresponding to $D_{I2}$) is
characterized by $R_2 = 915$ +/- 90 mm (compared to 1030 mm simulated), $\rho_2 = 0.53$, $AET_2 = 800$ mm, and
corresponds to 15 % of $V_O$. $T_8$, presenting an intermediate land-use characteristics with low slope in the
downstream part, has $R_8 = 730$ +/- 150 mm, $\rho_8 = 0.46$, $AET_8 = 850$ mm, and $K_8 = 16$ %. The main differences
between $T_2$ and $T_8$ are the topography and the soil characteristics: $T_2$ is located in the upstream part of the Mefou
catchment and presents hilly landscape with steep slopes; $T_8$ is located in the east side presenting lowland with
important wetland area and overflow during wet seasons. These differences explain the lower values of $R$ for $T_8$
in comparison with $T_2$. These results are comforted by observed runoff coefficient for the second wet season in
2017: $T_2$ have an observed runoff coefficient of 0.53 compared to 0.40 for $T_8$ despite a similar urbanization rate
(40 % for $T_8$ and 46 % for $T_2$). In the same category, $T_3$ ($D_{I4}$) presenting a most natural land-use (10 % of urbanized
area) than $T_2$ and $T_8$, is characterized by $R_3 = 715$ +/-75 mm (compared to 700 mm simulated), $\rho_3 = 0.43$,
$AET_3 = 905$ mm, and $K_3 = 18.5$ %.
For the fourth category, $T_4$ is not impacted by urbanization ($U < 1\%$). $T_4$ is located downstream with low
slope, with soil condition and topography favouring overflow, infiltration and evaporation. $T_4$ presents the lowest
$R_4 = 370$ +/- 50 mm, $\rho_4 = 0.24$, $AET_4 = 1290$ mm, and $K_4 = 9$ %.
For the Mefou at Nsimalen, we obtain by aggregation of the eight sub-catchments: $R_O = 660$ +/- 65 mm
(compared to 645 mm measured at the outlet), $\rho_O = 0.41$ and $AET_O = 990$ mm. These values are very near from
observed data with annual runoff measured of 645 mm.
These results on the eight target sub-catchments are compared to other studies in the Nyong basin (see
Supplementary Material, Sect. 4) on natural basins (Maréchal et al., 2011; Olivry, 1979; Lefèvre, 1966) and on
one urbanized basin of the Nyong basin. These studies led to comparable results of $\rho$ with $\rho = 0.24$ +/- 0.06 on
natural basins, and $\rho = 0.88$ on the urbanized of Odza (Ngoumdoum, 2013) on Nyong basin.

### 6.3    Reconstruction of historical runoff and prediction for scenarios of land-use

The past-period 1930-2017 simulated for targets and at Nsimalen are presented in Fig. 12. The grey envelope
curve represents the estimation of runoff uncertainties due to annual precipitation (+/- 10 %) and hydrological
index (+/- 15 %). Except for $T_4$ (downstream part of Mefou catchment) and $T_1$ (area controlled by the dam), all the
target sub-catchments present an increasing trend of runoff in earlier or later date. In a context of no annual
precipitation trends over the period 1930-2017, the urbanization development and forestry retreat for six of the
eight targets result in clear increasing of $R$. We notice that the impact on runoff is until now lesser for $T_3$ and $T_8$.
In order to quantify impacts of land-use changes, we calculated mean values $\bar{P}$, $\bar{R}$ and $\bar{\rho}$, 5th percentile $Q_5$ and
95th percentile $Q_{95}$ for one past period (1950-1980) and one recent period (1987-2017) (see Table 6). The mean
precipitation over the first period is about 75 mm higher (+ 5%) than for the second period with a similar standard
deviation (200 mm). Excluding $T_1$, controlled by the dam, we obtained on the period 1950-1980 $\bar{R}$ between 315
and 764 mm, $\bar{\rho}$ between 0.20 and 0.47, $Q_5$ between 200 and 503 mm and $Q_{95}$ between 502 and 1108 mm. On the





period 1987-2017, $\bar{R}$ varies between 297 and 1104 mm, $\bar{\rho}$ between 0.19 to 0.70, $Q_5$ between 150 and 690 mm and $Q_{95}$ between 508 and 1627 mm.

For the second category ($T_5$, $T_6$, $T_7$), an increasing trend of $R$ is observed very early among the whole period for $T_5$ and $T_6$ which are nowadays the most urbanized target of the Mefou catchment following by $T_7$. Between the two periods (1950-1980) and (1987-2017), $\bar{R}$ increases from 45 % for $T_5$ to 79 % for $T_6$, $\bar{\rho}$ increases between 50 % for $T_5$ to 85 % for $T_6$. $Q_5$ increases of 37 % for $T_5$ to 74 % for $T_7$ and $Q_{95}$ increases of 47 % for $T_5$ to 82 % for $T_6$.

For the third category ($T_2$, $T_3$, and $T_8$), classified as peri-urban catchments, sub-catchments are characterized by an increasing $R$ more delayed compared to catchments of the first category. Current changes are deeply modifying these catchments for the last decade and the urbanization processes will be certainly higher than on other sub-catchments in the near-future due to the current extension of the urban area. Between the two periods, $\bar{R}$ increases of 25 % for $T_3$ to 62 % for $T_8$. $Q_5$ increases of 30 % for $T_2$ and $T_8$ but decreases from of 26 % for $T_3$ whereas the $Q_{95}$ increases for the three catchments, from 48 % for $T_3$ to 69 % for $T_8$. Land-use changes were significant only since 2000 for $T_3$, and impact on $R$ is moderate compared to the two other catchments.

For the fourth category ($T_4$), $T_4$ was not impacted by urbanisation, $\bar{\rho}$ is unchanged (0.20) and differences of $\bar{R}$ are only driven by $\bar{P}$ difference of -5% (from 1625 mm to 1550 mm) and impact mainly low flows (-26 % for $Q_5$). No major changes were observed in this area until nowadays, but in development projects and urban area extension will certainly impact $T_4$ in near-future.

At the Mefou scale and for the same two periods, we observe an increase of $\bar{R}$ of 27% (from 409 to 518 mm), an increase of $\bar{\rho}$ of 31 % (from 0.25 to 0.33), an increase of $Q_{95}$ of 29% (from 650 to 840 mm) and nearly no changes for $Q_5$ (from 280 mm to 273 mm). The impact of land-use changes is clear on annual runoff but driest years are much less impacted than wettest years.

In order to quantify only the impact of land-use changes on annual runoff, we applied a constant precipitation of 1580 mm (mean precipitation over the period) for the period 1930-2017 (Fig. 13). Until 1980, the impact of land-use changes seems quite limited, with an increase of 110 mm of $R$ (+ 30%). However, between 1980 and 2017, the increase of $R$ and $\rho$ under the same precipitation condition seems quite relevant with an increase of 53% (from 455 to 700 mm and from 0.29 to 0.44). These hydrological changes are associated with a huge increasing of urban areas from 38 km² over the whole catchment to 130 km² over the same period.

Finally, the model was applied to simulate the impact of scenarios of land-use changes corresponding to predictions of $I$ for 2030 (Fig. 10). Results show for $P$ = 1580 mm, $R$ = 840 +/- 150 mm, $\rho$ = 0.53+/- 0.06 (Fig. 13). We observe an increase of $R$ and $\rho$ of 85 % (from 455 to 840 mm and from 0.29 to 0.53) between 1980 and 2030. Even if this scenario is fictive, it is quite reliable due to the dynamic of land-use changes observed these few decades and knowing the most recent projection of population of Yaoundé.

In tropical context, few studies evaluated the impact of land-use conversion from natural to urbanized area. Most of the studies quantified the impact of conversion from forest or shrub cover into cropland (Gessess et al., 2015; Yira et al., 2016) or forest regeneration (Beck et al., 2013).



More studies of annual runoff urbanization impacts have been lead in temperate or Mediterranean climate

(see Supplementary Material, Sect. 7), Braud et al. (2013) observed a significate increases of quick-flow and
decrease of inter-flow and base-flow in urbanization context on the Yzeron catchment (150 km²). Through
modelling, Beighley et al. (2003) estimated for Atascadera Creek ($P = 610$ mm) an increase of $R$ of more than
80 mm (+115 %) for 8 % to 38 % of urban area and an increases of $R$ of 150 mm (+215 %) from 8 % to 52 % of
urban area.


**7**     **Conclusion**

Urbanization impacts drastically the water cycle, and this phenomenon will intensify in the future for most

tropical regions due to huge population growth and rural exodus. These impacts are complex and not yet quantified
especially in tropical area presenting sparse hydrological data. This work is part of the theme *Panta Rhei* of the
*IAHS*, and aims to study the impact of land-use change, especially due to urbanization, on annual runoff on the
tropical mesoscale catchment of the Mefou, Yaoundé, Cameroon.

The methodology combines the processing of historical sparse hydrological data and a dedicated short-term

instrumentation (2017-2018) in order to get heterogeneous set of catchments in terms of land-use. Data analysis
shows that there is no significant trend on annual precipitation for the last 90 years. However, the analysis of
historical precipitation/runoff data on different catchments with different land-uses show an increase of the annual
runoff coefficient due to urbanization from 22 % on natural basins to 77 % on urbanized basins.

A simple semi-distributed annual runoff model was developed taking into account non-stationarity due to

land-use changes. The model is based on a similar approach as proposed by Ponce and Shetty (1995) and uses a
hydrological index $I$ characterizing soil, topography and land-use. From data analysis on donor catchments,
empirical rules were established to calculate $I$, and the model parameters. The model can be represented by simple
abacus giving relationships between the annual precipitation $P$, the annual runoff $R$ and the hydrological index $I$.
The non-stationarity of the model is characterized by the hydrological index $I$ which is time variable depending
on land-use evolution.

Applications were first done on target sub-catchments of the Mefou basin in order to calculate the annual

water balance for catchments with different land-uses. Results show that that the Mfoundi catchment, integrating
the three more urbanized sub-catchments, contributes near to 40 % of the Mefou catchment at Nsimalen despite
covering only 23 % of the basin. On the opposite, the natural sub-catchment $T_4$, not yet impacted by urbanization,
contributes to only 9 % but covers 18 % of the Mefou catchment. The second result is the reconstruction of
historical annual runoff from 1930-2017 for the Mefou catchment at Nsimalen with satisfying performance in a
poorly-gauged context ($RMSE = 99$ mm; $r^2 = 0.68$; $\bar{E} = 14.5$ %) compared to the commonly used stationary annual
model (GR1A). The mean values of $P, R, \rho, Q_5$ and $Q_{95}$ over the two periods of 30 years before and during the
urbanized processes show changes at both the sub-catchment scale and the whole Mefou scale: for a decrease of
about 50 % of the forest area and an increase from 8 % to 35 % of the urban area between 1980 and nowadays, we
observe an increase of 53 % of $R$ (and $\rho$) for the Mefou catchment at Nsimalen. The Future scenario of land-use



proposed leads to an increase of $R$ and $\rho$ of 85 % between 1980 and 2030. The observed and simulated values for
heterogeneous land-use characteristics are in line with other studies which observed an increasing of annual runoff
for catchment faced to urbanization (Braud et al., 2013; White and Greer, 2006; Barron et al., 2003).
The coupled experimental-modelling approach proposed herein opens promising perspectives regarding the
evaluation of annual runoff in catchments under changes. Nowadays, development of low cost monitoring sensors
and crowdsourced sciences open opportunities to get more easily data to calibrate and validate models. In changing
context, the development of coupled non-stationary modelling and dedicated instrumentation can be useful to
improve the capability of stakeholders to make predictions of the hydrological dynamics of tropical peri-urban
catchments.






*Data availability.* Please contact the first author (Camille Jourdan) for data.

*Author contributions.* C.J, V.B.E, RM., D.S., and E.S. developed the study framework and prepared the paper
structure; C.L. elaborated a first; C.J., A.F., J.B.B., V.B.E., D.S., B.N.G., J.R.N., and S.V.E performed field
instrumentation and measurements; A.C, F.C contributed to the interpretation and the discussion of the results;
C.J. and R.M. wrote the paper with contributions from all authors.

*Supplement.* A supplementary material is provided to complete the paper.

*Competing interests.* The authors declare that they have no conflict of interest.

*Acknowledgements :*We acknowledge the LMI PICASS'EAU and the LMI DYCOFAC of the French National
Research Institute for Development (IRD) for their support; Dr J-C Ntonga (IRGM-CRH) for his collaboration ;
Jean-Pierre Bricquet, Hélène Mathieu-Subias for technical support in field instrumentation and data collection;
Marielle Gosset for the financial support; David Badoga, Moustapha Djangue, Lionel Yossa, Souleyman Abba,
Daoud Nsangou and all the students involved in the field schools Hydraride (2016 and 2017) for their contribution;
and Nathalie Rouché for historical database information (SIEREM). Authors have a thought for the late Jean-
Pierre Bedimo Bedimo for whom it was among his last projects.




**Figures**






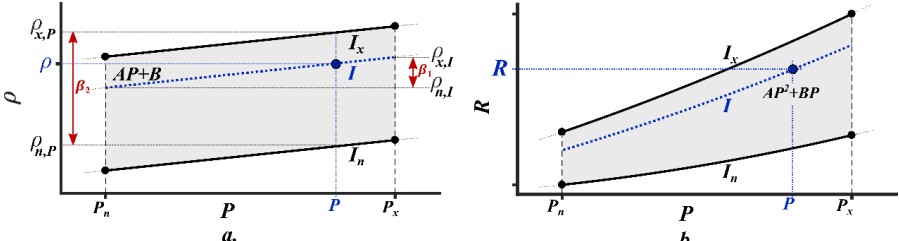


**Figure 1.** The model general relationships between annual precipitation $P$, annual runoff $R$, annual runoff coefficient $\rho$ and the hydrological index $I$. (a) Relationship between $\rho$ and $P$ with $\rho = AP+B$ for different ranges of $I$; (b) Relationship between $R$ and $P$ with $R = AP^2+BP$ for different ranges of $I$. $P_n$: minimum $P$; $P_x$: maximum $P$; $I_n$: minimum $I$; $I_x$: maximum $I$. The point in each graph is an example of annual runoff coefficient and annual runoff estimation for a precipitation $P$ and an hydrological index $I$.







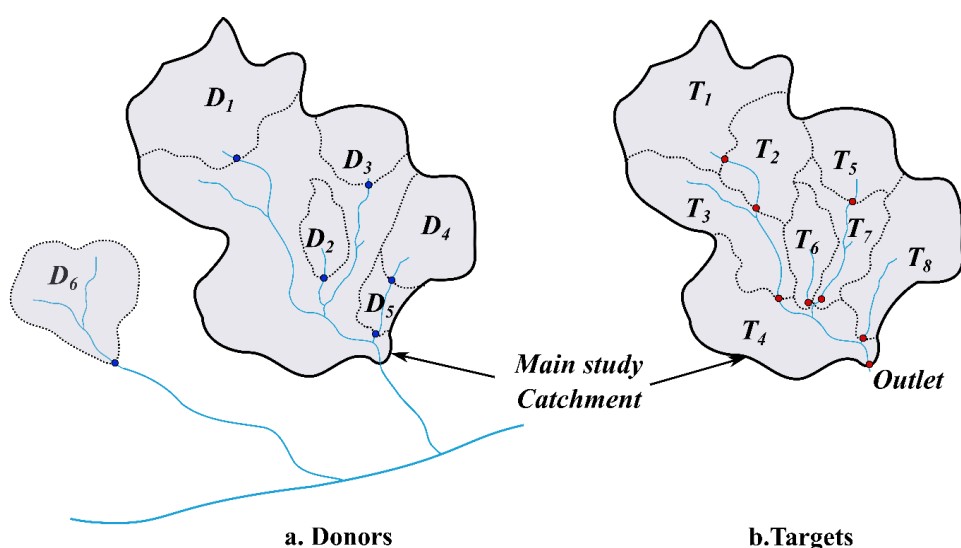


**Figure 2.** Example of fictive donor (a) and target (b) catchments.





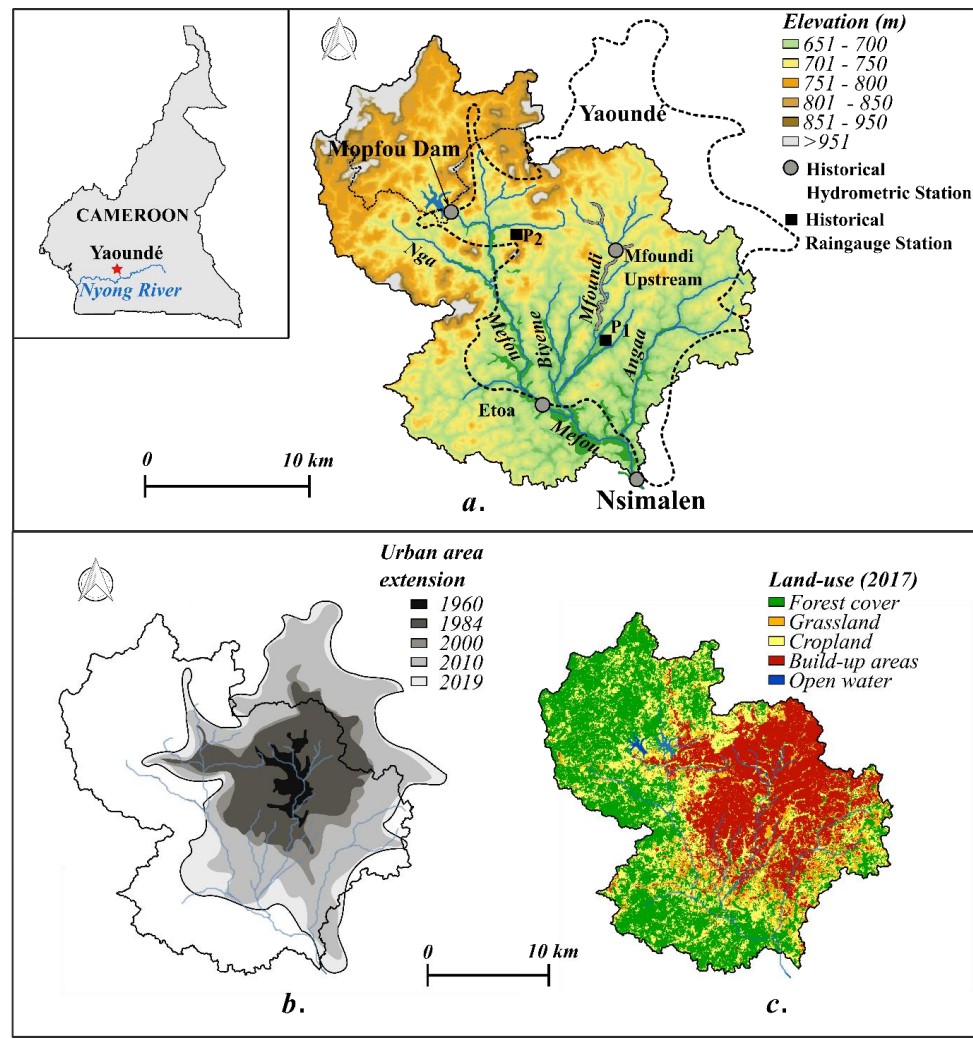


**Figure 3.** The Mefou catchment at Nsimalen: (a) Location, channel network and topography. (b) Urban areas
evolution from 1956 to 2018 from historical photography (Moffo, 2011) and satellite images (Google Earth ®) (c)
Land-use extracted from the product Land Cover for Africa of European Space Agency (ESA-LC), based on one
year Sentinel-2A observations from December 2015 to December 2016.







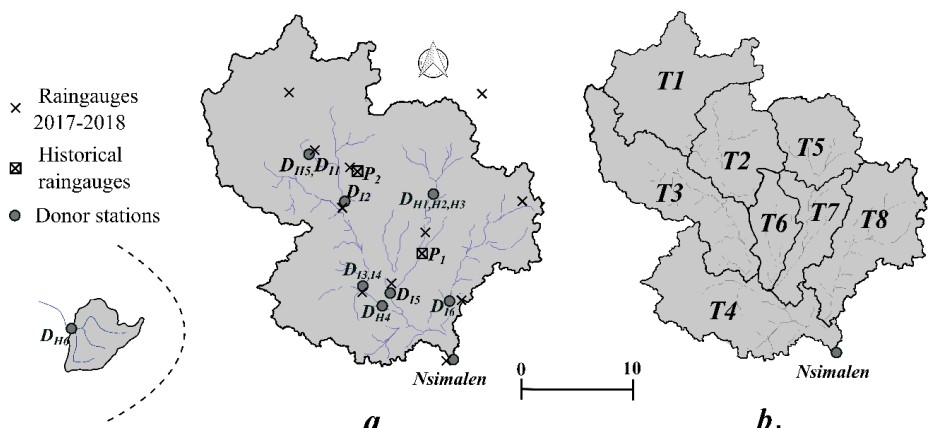

**Figure 4.** The Mefou catchment: (a) Location of raingauges and limnimetric stations on donor catchments (NB: the donor $D_{H6}$ is located outside of the Mefou basin at 40 km south-west); (b) target catchments.




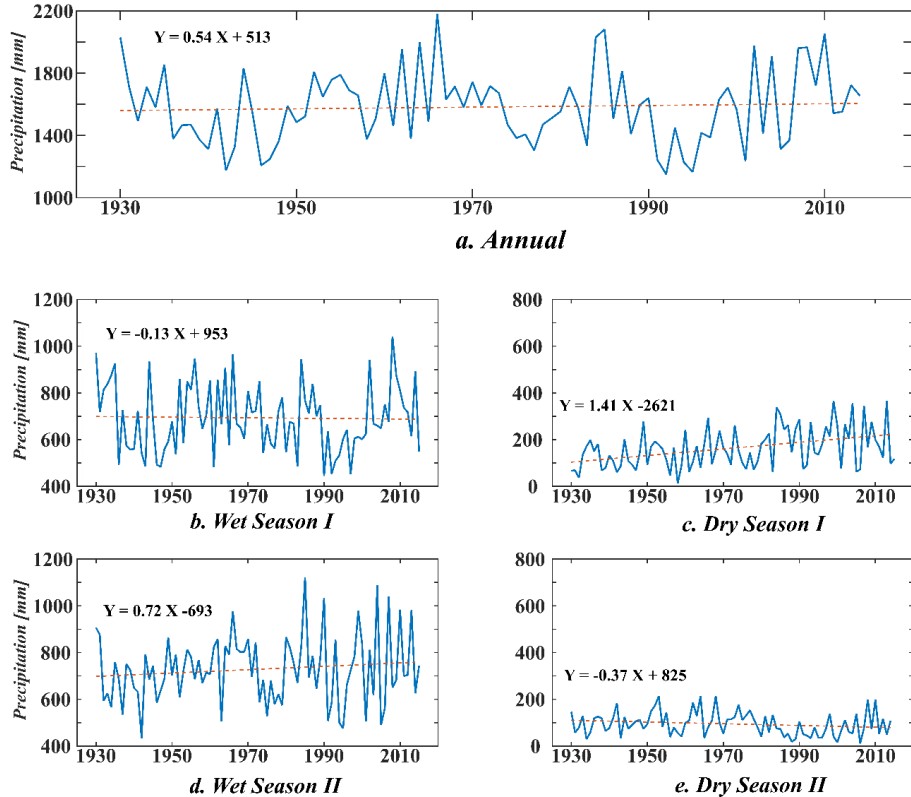

**Figure 5.** Annual and seasonal precipitation: (a) Annual precipitation from 1930 to 2015 from historical raingauge $P_1$ (Mvan Airport). (b) Precipitation during the wet season *I* March to June. (c) Precipitation during the dry season *I* July to August. (d) Precipitation during the wet season *II* September to November. (e) Precipitation during dry season *II* December to February.



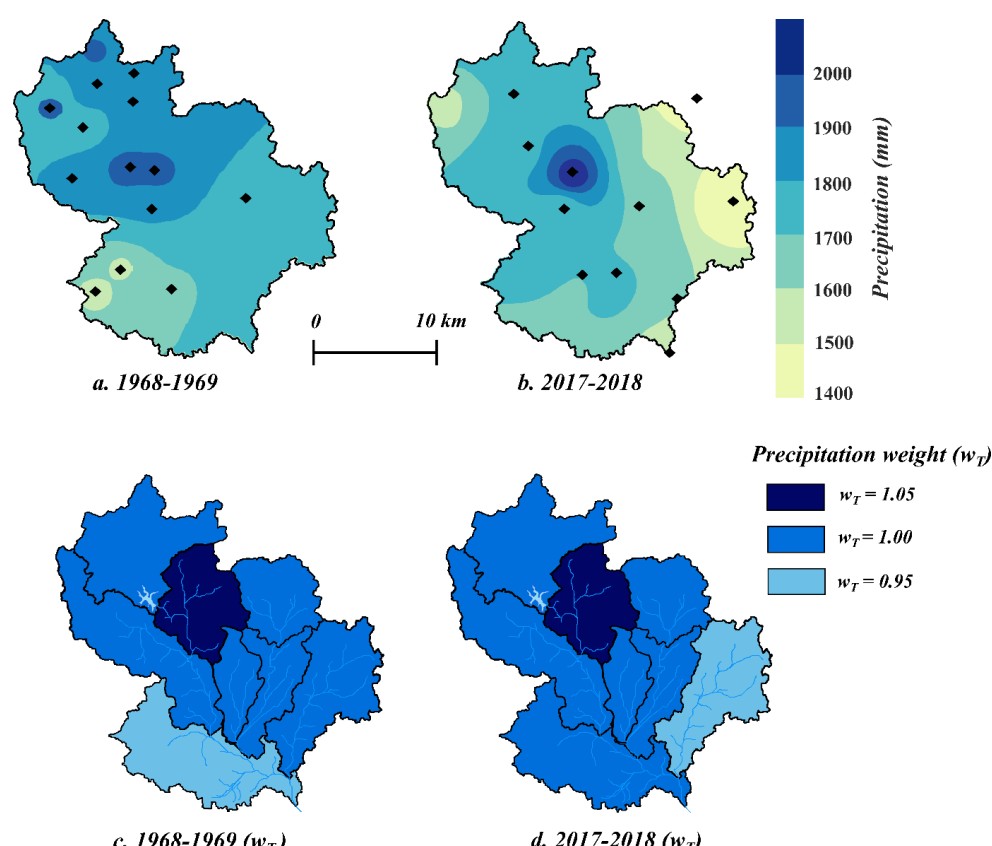

**Figure 6.** Annual precipitation map on the Mefou catchment: (a) for 1968-1969 (Ikounga, 1978); (b) for the instrumented period (March 2017 to February 2018). Precipitation weights on the 8 target catchments: (c) calculated from (a); (d) calculated from (b).





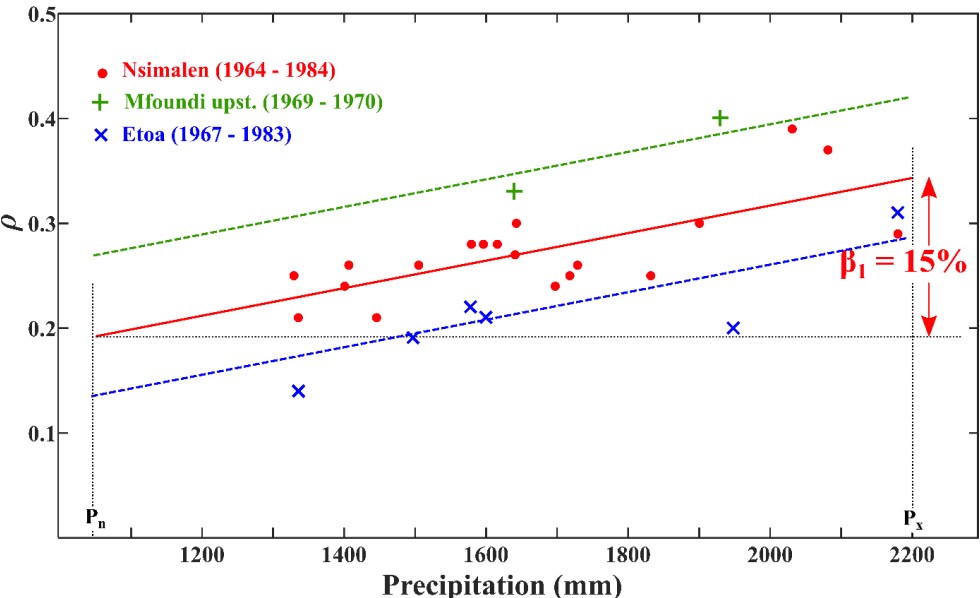


**Figure 7.** Relationship between the annual runoff coefficient $\rho$ and the annual precipitation $P$ on three stations
(Nsimalen, Mfoundi and Etoa; see the location on Fig. 3.a) for periods before 1985 with low impact of
urbanization.





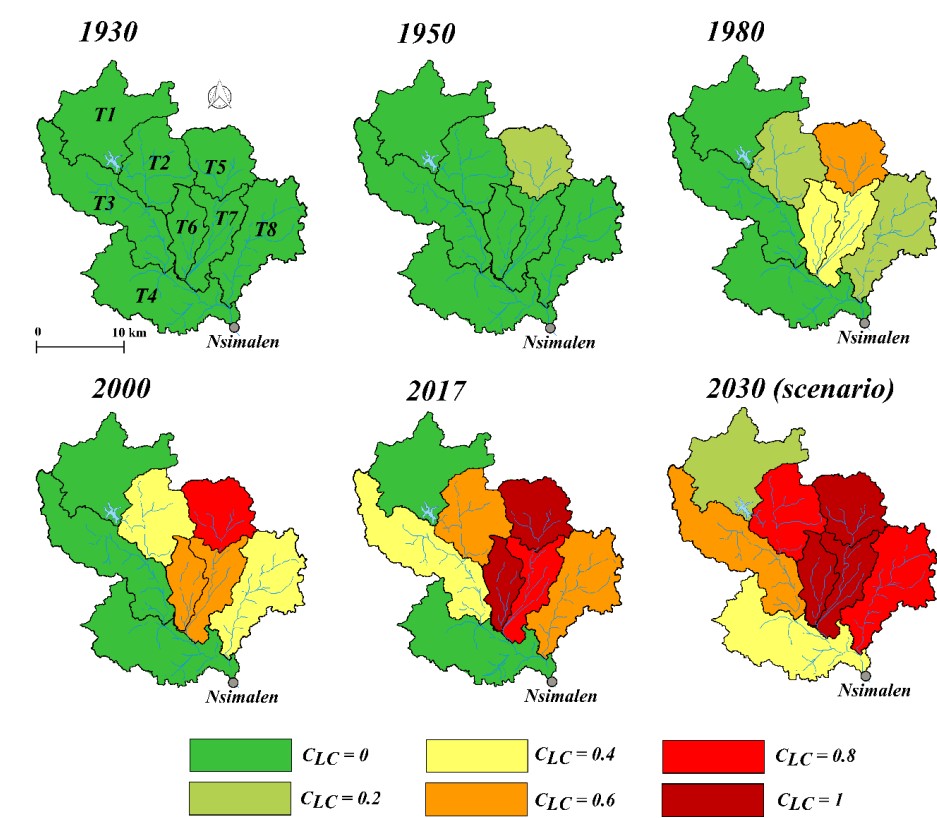


**Figure 8.** Evolution of the component $C_{LC}$ (see Section 5.1) of the hydrological index $I$ over the target catchments
for 1930, 1950, 1980, 2000, 2017 and scenario for 2030 (see references of land-use sources in Table 2).






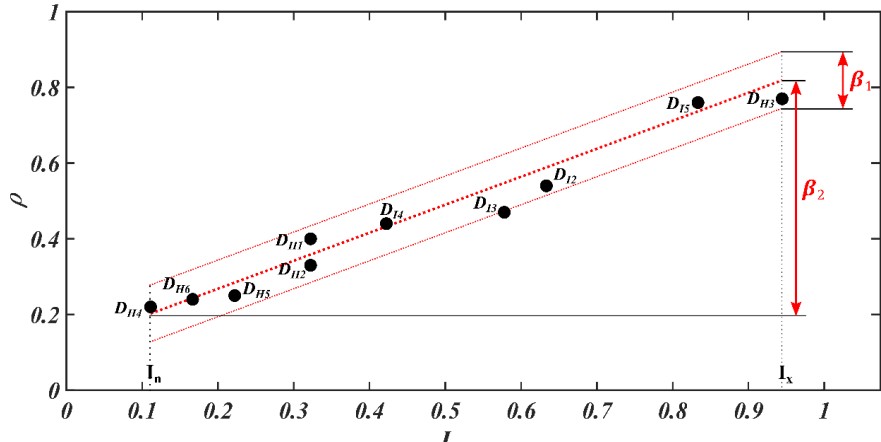


**Figure 9.** Linear relationship between the hydrological index $I$ and the annual runoff coefficient $\rho$ based on donor
catchments (Table 1). The term $\beta_1$ represents the variation of $\rho$ for a given value of $I$ and for a large range of
precipitation $P$ with $P_n < P < P_x$. The term $\beta_2$ represents the variation of $\rho$ for a given value of $P$ and for a large
range of $I$ with $I_n < I < I_x$.






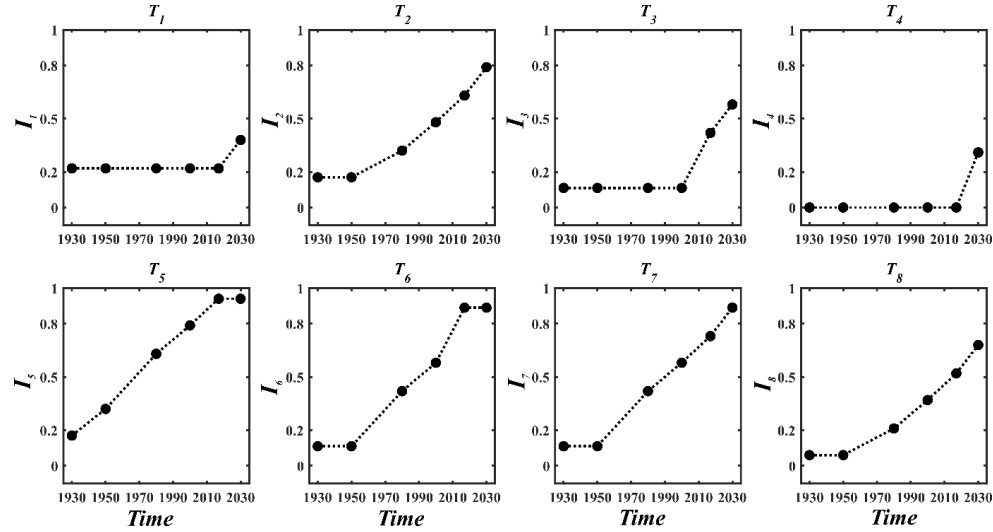

**Figure 10.** Evolution of the hydrological index *I* over time for target catchments for the period 1930 - 2030. The
points correspond to the dates where references of land-use sources are available (Table 2 and Fig. 8).





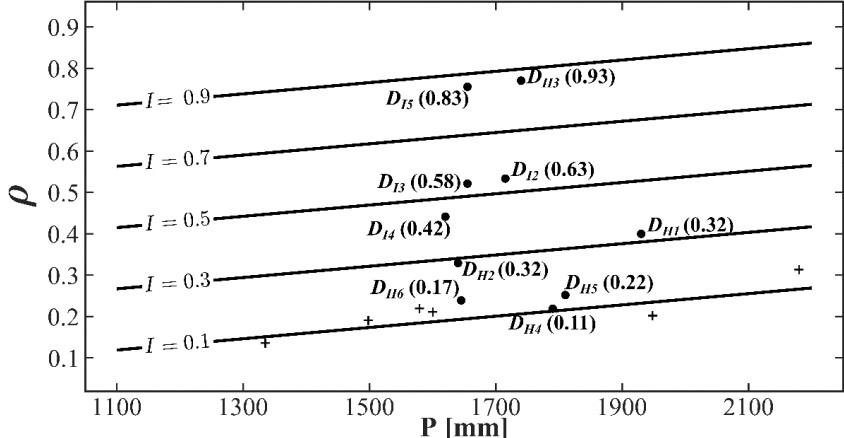


**Figure 11.** Abacus obtained from the model calibrated parameters for the relationship between annual runoff coefficient $\rho$ and the annual precipitation $P$ for different values of the hydrological index $I$. Each point corresponds to donor catchment; the value of the calculated hydrological index $I$ (Table 3) is indicated into brackets. Crosses (+) correspond to available information at Etoa station between 1967 and 1983 for an unchanged hydrological index $I$ estimated to 0.11.



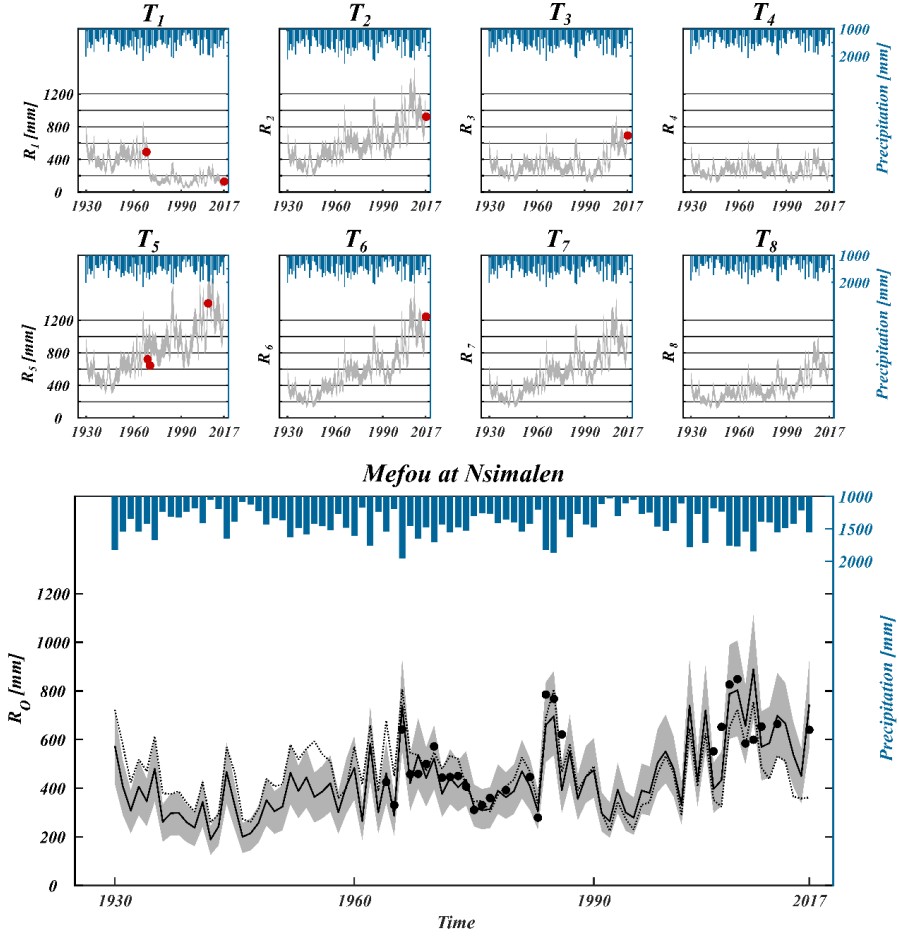

**Figure 12.** Annual runoff simulated on the 8 target catchments ($T_1$ to $T_8$) and on the Mefou catchment at Nsimalen
for the period 1930-2017 (dark line with grey uncertainty range due to precipitation +/- 10 % and hydrological
index $I$ +/- 15 % estimation). The black points indicate the observed values on Nsimalen. The simulation with
*GR1A* is presented with a dashed line.



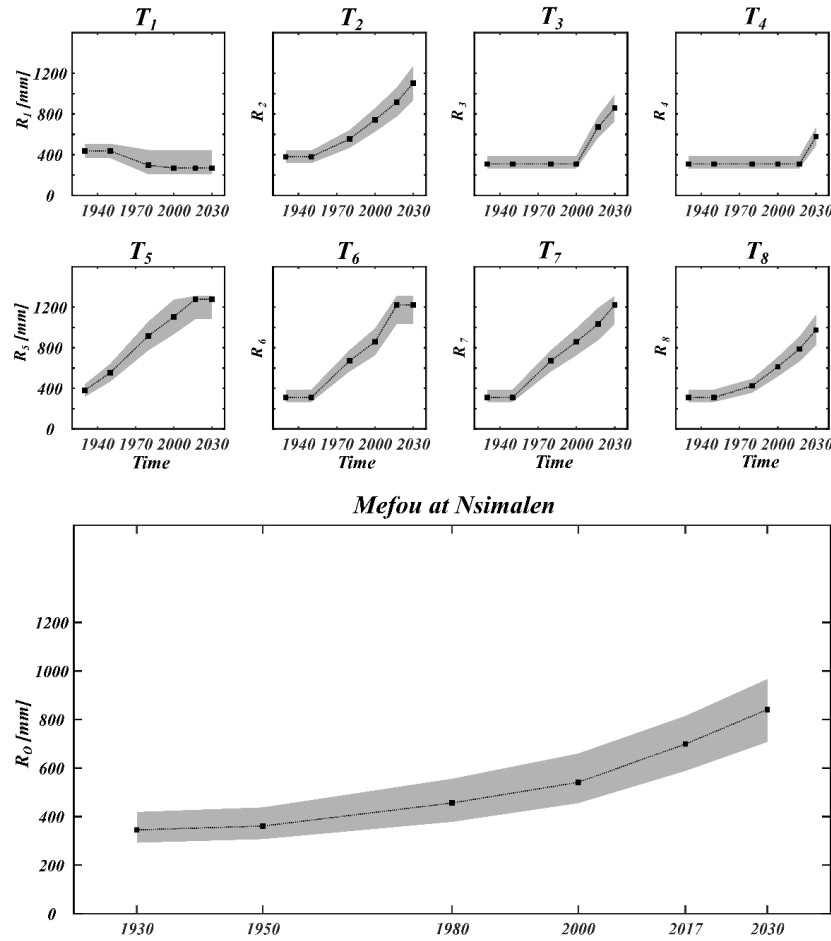

**Figure 13.** Simulated annual runoff for the target catchments ($T_1$ to $T_8$) and on the Mefou catchment at Nsimalen
for a reference precipitation $P_R = (P_n + P_x) / 2$ over the period 1930 – 2030 (dark line with grey uncertainty range
due to the hydrological index $I$ +/- 15 % estimation).

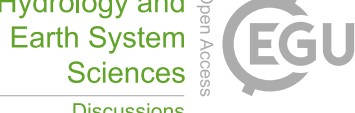

**Tables**



**Table 1.** Characteristics of the donor catchments: name, reference, hydrometric station name, area, period of observation, annual precipitation $P$, annual runoff $R$, annual evapotranspiration $AET = P - R$, annual runoff coefficient $\rho$, the slope index $S_l$, the proportion of hydromorphic soil $HS$, and the proportion of urban areas $U$.

| Name | Reference | Hydrometric station | Area [km²] | Period | P [mm] | R [mm] | AET [mm] | $\rho$ | $S_l$ [%] | HS [%] | U [%] |
|---|---|---|---|---|---|---|---|---|---|---|---|
| $D_{H1}$ | Srang (1972) | MFOUNDI UPST | 40 | 1970-71 | 1930 | 772 | 1158 | 0.40 | 11.5 | 0 | 5 |
| $D_{H2}$ | Srang (1972) | MFOUNDI UPST | 40 | 1971-72 | 1640 | 540 | 1100 | 0.33 | 11.5 | 0 | 5 |
| $D_{H3}$ | Nguemou (2008) | MFOUNDI UPST | 40 | 2006-07 | 1740 | 1340 | 400 | 0.77 | 11.5 | 0 | > 75 |
| $D_{H4}$ | SNEC (1969) | ETOA | 235 | 1968-69 | 1790 | 392 | 1398 | 0.22 | 9.0 | 44 | <1 |
| $D_{H5}$ | SNEC (1969) | MOPFOU | 70 | 1968-69 | 1810 | 456 | 1354 | 0.25 | 13.5 | 0 | <1 |
| $D_{H6}$ | Ikounga (1978) | S3 | 24 | 1974-77 | 1645 | 393 | 1252 | 0.24 | 13.4 | >5 | <1 |
| $D_{I1}$ | | MOPFOU | 70 | 2017-18 | 1650 | 100 | - | 0.06 | 13.5 | 0 | <1 |
| $D_{I2}$ | | MEFOU UPST | 47 | 2017-18 | 1715 | 915 | 800 | 0.54 (0.53*) | 13.6 | 3 | 46 |
| $D_{I3}$ | Dedicated short-term instrumentation | CANABOIS | 120 | 2017-18 | 1655 | 863 | 792 | 0.47 | 10.0 | 6 | 28 |
| $D_{I4}$ | | CANA-NKOM | 73 | 2017-18 | 1620 | 712 | 908 | 0.44 | 10.0 | 8 | 9 |
| $D_{I5}$ | | ECOPARK | 21 | 2017-18 | 1655 | 1250 | 405 | 0.76 | 9.8 | 35 | 87 |
| $D_{I6}$ | | ANGAA* | 54 | 2017-18 | 575* | 230* | - | 0.40* | 6.7 | 24 | 45 |

* The data on the Angaa catchment ($D_{I6}$) covers only the period from September to December. For comparison, the value of $\rho$ is given for the same period for the Mefou Upstream ($D_{I2}$) which is characterized by similar value of $U$.



**Table 2.** Reference studies, global products and resolution used to estimate each component ($C_T$, $C_S$ and $C_{LC}$) of
the index hydrological index *I* for 1930, 1950, 1980, 2000, 2017 and 2030. SRTM: Shuttle Radar Topography
Mission 2014; *ESA CCI LC*: European Space Agency Climate Change Initiative Land Cover.

| Type | Period | References | Product | Spatial resolution |
|---|---|---|---|---|
| *Topography* | *1930 - 2030* | SNEC (1969)<br>Srang (1972)<br>Ikounga (1978)<br>Olivry (1979) | *SRTM 2014* | *30 m x 30 m* |
| *Soil* | *1930 - 2030* | Bachelier (1959)<br>Humbel and Pellier (1969)<br>Pellier (1969) | *SRTM 2014* | *30 m x 30 m* |
| *Land-use* | *1930* | Franqueville (1968, 1979) | *Population estimation* | - |
| | *1950* | Moffo (2016) | *Aerial photography* | - |
| | *1980* | Ebodé (2017) | *Landsat image* | - |
| | | CUY (2008) | *Urban area map in CUY (2008)* | - |
| | *2000* | Midekisa et al. (2017)<br>Moffo (2017) | *Map in Midekisa et al. (2017)* | *300 m x 300 m* |
| | *2017* | Midekisa et al. (2017) | *Map in Midekisa et al. (2017)* | *300 m x 300 m* |
| | | Ebodé (2017) | *Landsat image* | - |
| | *2030* | UNDESA (2017) | *Prediction of population growth in UNDESA (2017)* | *By city and country* |







**Table 3.** Characteristics of the three components $C_T$, $C_S$, $C_{LC}$, and the hydrological index $I$ of the 12 donor
catchments (Fig. 4b).

| Name | $C_T$ | $C_S$ | $C_{LC}$ | $I$ |
|------|-------|-------|----------|-----|
| $D_{H1}$ | 0.50 | 1.00 | 0.20 | 0.32 |
| $D_{H2}$ | 0.50 | 1.00 | 0.20 | 0.32 |
| $D_{H3}$ | 0.50 | 1.00 | 1.00 | 0.94 |
| $D_{H4}$ | 0.50 | 0.50 | 0.00 | 0.11 |
| $D_{H5}$ | 1.00 | 1.00 | 0.00 | 0.22 |
| $D_{H6}$ | 1.00 | 0.50 | 0.00 | 0.17 |
| | | | | |
| $D_{I1}$ | 1.00 | 1.00 | 0.00 | 0.22 |
| $D_{I2}$ | 0.50 | 1.00 | 0.60 | 0.63 |
| $D_{I3}$ | 0.50 | 0.50 | 0.60 | 0.58 |
| $D_{I4}$ | 0.50 | 0.50 | 0.40 | 0.42 |
| $D_{I5}$ | 0.50 | 0.00 | 1.00 | 0.83 |
| $D_{I6}$ | 0.00 | 0.00 | 0.60 | 0.47 |






**Table 4.** Characteristics of the 8 target catchments (Fig. 4b): area, the two components $C_T$, $C_S$ and the dynamic
time variable component $C_{LC}$ given in Fig. 8 (see references of land-use sources in Table 2).

| Name | Area [km²] | $C_T$ | $C_S$ | $C_{LC}$ | | | | | |
|---|---|---|---|---|---|---|---|---|---|
| | | | | 1930 | 1950 | 1980 | 2000 | 2017 | 2030 |
| $T_1$ | 70 | 1.0 | 1.0 | 0.00 | 0.00 | 0.00 | 0.00 | 0.00 | 0.20 |
| $T_2$ | 47 | 1.0 | 1.0 | 0.00 | 0.00 | 0.20 | 0.40 | 0.60 | 0.80 |
| $T_3$ | 73 | 0.5 | 0.5 | 0.00 | 0.00 | 0.00 | 0.00 | 0.40 | 0.60 |
| $T_4$ | 75 | 0.0 | 0.0 | 0.00 | 0.00 | 0.00 | 0.00 | 0.00 | 0.40 |
| $T_5$ | 40 | 0.5 | 1.0 | 0.00 | 0.20 | 0.60 | 0.80 | 1.00 | 1.00 |
| $T_6$ | 24 | 0.5 | 0.0 | 0.00 | 0.00 | 0.40 | 0.60 | 1.00 | 1.00 |
| $T_7$ | 30 | 0.5 | 0.0 | 0.00 | 0.00 | 0.40 | 0.60 | 0.80 | 1.00 |
| $T_8$ | 62 | 0.0 | 0.0 | 0.00 | 0.00 | 0.20 | 0.40 | 0.60 | 0.80 |





**Table 5**. Calculated values (either from observation or from model simulation as indicated) of $R$, $\rho$, $AET$, and $K$
for each target catchments and for the whole Mefou catchment at Nsimalen taking into account +/- 10 %
uncertainty on $P$ and +/- 15 % $I$ for the hydrological year 2017-2018.

| Catch. Name | Method | P [mm] | R [mm] | AET [mm] | $\rho$ | K [%] |
|:---:|:---:|:---:|:---:|:---:|:---:|:---:|
| $T_1$ | Obs. ($D_{I1}$) | 1500 | 100 ± 25 | - | - | 2.5 |
| $T_2$ | Obs. ($D_{I2}$) | 1715 | 915 ± 90 | 800 | 0.53 | 15.0 |
| $T_3$ | Obs. ($D_{I3}$) | 1620 | 715 ± 75 | 905 | 0.43 | 18.5 |
| $T_4$ | Sim. | 1660 | 370 ± 75 | 1290 | 0.24 | 9.0 |
| $T_5$ | Sim. | 1620 | 1230 ± 125 | 390 | 0.76 | 18.0 |
| $T_6$ | Obs. ($D_{I5}$) | 1655 | 1130 ± 125 | 525 | 0.68 | 10.0 |
| $T_7$ | Sim. | 1650 | 1030 ± 200 | 620 | 0.62 | 11.0 |
| $T_8$ | Sim. | 1580 | 730 ± 150 | 775 | 0.46 | 16.0 |
| **Mefou** | Sum of volumes of $T_i$ | 1650 | 660 ± 65 | 990 | 0.41 | 100 |






**Table 6.** Mean values of $P$, $R$, $\rho$, and percentile $Q_{95}$ and $Q_5$ for 1950-1980 and 1987–2017 for the eight sub-catchments and the whole catchment.

| Name | 1950 - 1980 | | | | | 1987 - 2017 | | | | | Changes | | | | |
|---|---|---|---|---|---|---|---|---|---|---|---|---|---|---|---|
| | $\bar{P}$ [mm] | $\bar{R}$ [mm] | $\bar{\rho}$ | $Q_{95}$ [mm] | $Q_5$ [mm] | $\bar{P}$ [mm] | $\bar{R}$ [mm] | $\bar{\rho}$ | $Q_{95}$ [mm] | $Q_5$ [mm] | $\bar{P}$ [%] | $\bar{R}$ [%] | $\bar{\rho}$ [%] | $Q_{95}$ [%] | $Q_5$ [%] |
| $T_1$ | - | 356 | - | - | - | - | 144 | - | - | - | - | - | - | - | - |
| $T_2$ | 1706 | 529 | 0.31 | 803 | 346 | 1628 | 803 | 0.48 | 1255 | 454 | - 5 | +52 | +58 | +56 | +31 |
| $T_3$ | 1625 | 324 | 0.20 | 528 | 200 | 1550 | 405 | 0.25 | 783 | 149 | - 5 | +25 | +28 | +48 | -26 |
| $T_4$ | 1625 | 324 | 0.20 | 528 | 200 | 1550 | 297 | 0.19 | 508 | 149 | - 5 | -8 | -6 | -4 | -26 |
| $T_5$ | 1625 | 764 | 0.47 | 1108 | 503 | 1550 | 1104 | 0.70 | 1627 | 690 | - 5 | +45 | +50 | +47 | +37 |
| $T_6$ | 1625 | 477 | 0.29 | 743 | 276 | 1550 | 852 | 0.54 | 1352 | 461 | - 5 | +79 | +85 | +82 | +67 |
| $T_7$ | 1625 | 434 | 0.27 | 678 | 247 | 1550 | 741 | 0.47 | 1143 | 429 | - 5 | +71 | +77 | +69 | +74 |
| $T_8$ | 1544 | 315 | 0.20 | 502 | 197 | 1473 | 511 | 0.34 | 846 | 256 | - 5 | +62 | +69 | +69 | +30 |
| $Mefou$ | 1625 | 409 | 0.25 | 650 | 280 | 1550 | 518 | 0.33 | 840 | 273 | -5 | +27 | +31 | +29 | -2 |







**Appendix A**

**List of Notations**
$A$ : Parameter of the equation: $\rho = AP + B$ [ - ]
$A_i$ : Area of target sub-catchment $i$ [ L² ]
$A_O$ : Area of the whole catchment [ L² ]
$a$ : Parameter of the model in $\rho_D = aI + b$ [ - ]
$AET$ : Annual evapotranspiration [ L ]
$B$ : Parameter of the equation: $\rho = AP + B$ [ - ]
$b$ : Second parameter of the model in $\rho_D = aI + b$ [ - ]
$C_i$ : Component $i$ of the hydrological index [ - ]
$C_{LC}$ : Land-use component of the hydrological index $I$ [ - ]
$C_S$ : Soil component of the hydrological index $I$ [ - ]
$C_T$ : Topographic component of the hydrological index $I$ [ - ]
$CN$ : Curve Number in SCS method
$D$ : Donor catchments [ - ]
$D_{Ii}$ : Donor catchment $i$ from dedicated short-term instrumentation [ - ]
$D_{Hi}$ : Donor catchment $i$ from historical database [ - ]
$E$ : Normalized Error (Eq. 10) [ - ]
$\bar{E}$ : Mean Absolute Normalized Error (Eq. 11) [ - ]
$G$ : Term defined in Eq 16 [ - ]
$HS$ : Proportion of hydromorphic soil [ - ]
$I$ : Hydrological index [ - ]
$I_n$ : Maximum value of I [ - ]
$I_x$ : Minimum value of I [ - ]
$K_i$ : Contribution of sub-catchment $i$ to the whole catchment ($K_i = V_i / V_O$) [ - ]





| 854 | $P$ | : | Annual precipitation [L] |
|---|---|---|---|
| 855 | $P_m$ | : | Mean annual precipitation over the historical database [ L] |
| 856 | $P_n$ | : | Minimal annual precipitation over the historical database [L] |
| 857 | $P_R$ | : | Reference annual precipitation corresponding to $P_R = \frac{P_x + P_n}{2}$ [L] |
| 858 | $P_{Ti}$ | : | Annual precipitation of target catchment $i$ (Eq. 12) [L] |
| 859 | $P_x$ | : | Maximal annual precipitation over the historical database [L] |
| 860 | $\bar{P}$ | : | Mean annual precipitation for a period of 30 years (1950-1980 and 1987-2017) |
| 861 | $P_1$ | : | Annual precipitation at historical raingauge $P_1$ [ L ] |
| 862 | $P_2$ | : | Annual precipitation at historical raingauge $P_2$ [ L ] |
| 863 | $PET$ | : | Potential evapotranspiration [L] |
| 864 | $Q_5$ | : | Annual runoff corresponding to the 5[th]-percentile over a period of 30 years [ L ] |
| 865 | $Q_{95}$ | : | Annual runoff corresponding to the 95[th]-percentile over a period of 30 years [ L ] |
| 866 | $R$ | : | Annual Runoff [ L ] |
| 867 | $R_i$ | : | Annual Runoff of target sub-catchment $i$ [ L ] |
| 868 | $R_O$ | : | Annual Runoff of the whole catchment [ L ] |
| 869 | $\bar{R}$ | : | Mean annual simulated runoff for a period of 30 years (1950-1980 and 1987-2017) |
| 870 | $r^2$ | : | Coefficient of determination (Eq. 9) |
| 871 | $RMSE$ | : | Roots Mean Square Error [ L ] |
| 872 | $S_i$ | : | Slope index [ % ] |
| 873 | $T_i$ | : | Target sub-catchment $i$ [ - ] |
| 874 | $U$ | : | Proportion of urbanized area [ % ] |
| 875 | $V_i$ | : | Annual volume at sub-catchment $i$ [L$^3$] |
| 876 | $V_O$ | : | Annual volume at the whole catchment [L$^3$] |
| 877 | | | |
| 878 | $\beta_1$ | : | Parameter of the model ($\beta_1 = \rho_{x,I} - \rho_{n,I}$) [ - ] |
| 879 | $\beta_2$ | : | Constant characteristic ($\beta_2 = \rho_{x,P} - \rho_{n,P}$) |
| 880 | $\rho$ | : | Annual runoff coefficient [ - ] |



| 881 | $\bar{\rho}$ | : | Mean simulated annual runoff coefficient for a period of 30 years (1950-1980 and |
| 882 | | | 1987-2017) [ - ] |
| 883 | $\rho_D$ | : | Simulated annual runoff coefficient from donors regression: $\rho_D = aI + b$ [ - ] |
| 884 | $\rho_{n,I}$ | : | $\rho$ for $P_n$ and $I$ [ - ] |
| 885 | $\rho_{n,P}$ | : | $\rho$ for $I_n$ and $P$ [ - ] |
| 886 | $\rho_{x,I}$ | : | $\rho$ for $P_x$ and $I$ [ - ] |
| 887 | $\rho_{x,P}$ | : | $\rho$ for $I_x$ and $P$ [ - ] |
| 888 | $\omega_i$ | : | Weight attributed to the component i of the hydrological index in Eq. 7 [ - ] |
| 889 | $\omega_{LC}$ | : | Weight attributed to the land-use component of the hydrological index [ - ] |
| 890 | $\omega_S$ | : | Weight attributed to the soil component of the hydrological index [ - ] |
| 891 | $\omega_T$ | : | Weight attributed to the topographic component of the hydrological index [ - ] |
| 892 | | | |
| 893 | | | |

**List of abbreviations**

| 895 | *CUY* | : | Communauté Urbaine de Yaoundé |
| 896 | *ESA CCI LC* | : | European Space Agency Climate Change Initiative |
| 897 | *IAHS* | : | International Association of Hydrological Sciences |
| 898 | *LULC* | : | Land-Use Land-Cover |
| 899 | *PUB* | : | Prediction in Ungauged Basins |
| 900 | *SCS* | : | Soil Conservation Services |
| 901 | *SNEC* | : | Société Nationale des Eaux du Cameroun |
| 902 | *SRTM* | : | Shuttle Radar Topography Mission |
| 903 | *UNDESA* | : | United Nations Department of Economic and Social Affairs |
| 904 | | | |
| 905 | | | |





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
