# Peer review of "A non-stationary model for reconstruction of historical annual runoff on tropical catchments under increasing urbanization (Yaoundé, Cameroon)"

_Hydrology and Earth System Sciences, 2019_

## Referee Comment (RC1) · Anonymous Referee #1 · 22 May 2019

The paper develops and simple regression-based annual runoff model and evaluates the model's performance in data sparse environments in Cameroon to assess the role of urbanization on runoff. The model uses and catchment index I, similar to the CN of the SCS-CN equation. The difference was that this index was designed for annual runoff. I was not able to find any new hypotheses, concepts, science questions and advancements in this paper. I'd would have appreciated if the authors at least used the Budyko framework or simple annual water models like the ABC model, both of which has more sound conceptual basis that involve water and energy. I'm not convinced that this paper deserves publication in HESS. It looks like a nice project report.

---

## Referee Comment (RC2) · Anonymous Referee #2 · 30 Jun 2019

The scope of this paper is to develop a combined approach of data acquisition and the development of a new semi-distributed model taking into account land-use changes to reconstruct and predict annual runoff on a catchment exposed to high urban increase. The research question is interesting, and the methods used in this study are well explained. However, I have some major comments listed below. 1. Lines 122-123: How about potential evaporation? The effect of potential evaporation on annual runoff may need to be incorporated into the model. 2. Line 139 equation (1): how is this equation obtained from SCS curve number method applied to the annual scale? 3. The effect of inter-annual soil water storage carry-over is not considered. As shown in Lines 357-358, there are humid (1960-1970, 1980-1990, and 2006-2013) and dry (1935-1950,
1970-1980, and 1990-2000) periods. The storage carry-over from the transition between wet and dry periods may be not negligible. 4. Lines 358-362: The change of seasonal rainfall pattern will affect the annual runoff. It will be good to explore or discuss the potential impact. 5. Figure 7: Considering the annual runoff coefficient, the uncertainty for the linear model between runoff coefficient and precipitation is large (i.e., the scatter of data points around the line). This is due to the following potential reasons: 1) the validity of equation (1); 2) the effect of potential evaporation; 3) the effect of inter-annual soil water storage change; and 4) the effect of seasonal variation of precipitation. 6. In this paper, the temporal change of I is used to model the effect of land use change. However, the land use change may be better to be reflected in the parameters of the rainfall-runoff relation (e.g., A and B parameters). What are the estimated values of A and B? The parameters A and B have physical meanings. For example, soil water storage capacity is an important parameter in the SCS curve number method, and it is a function of land cover and land use. Land use change will cause the change of soil water storage capacity. Some minor comments: Lines 29 and 117: $km^2$? Same for other places. Line 30: change "an" to "a" The Abstract section could be shortened. Line 59: change "anthropic" to "anthropogenic"? Line 72: delete "people" Line 95: change "as" to "of"? Line 103: change "recently" to "recent" Line 124: "short-terms" to "short-term"? Line 139: $AP^2$ Line 175 and Figure 1: Why is $\beta_2$ similar for all values of P? Why are the upper bound and lower bound of Figure 1a in parallel? Line 396: Px=200mm? May be a typo. Line 433: change "(U" into "U" Line 461: "quite clear in term of runoff with for D14 runoff value up to 160 mm..." There is a typo, it should be D12 not D14. Line 499: A " " is missing after "consequently". Line 531: There are 12 donors totally, but only 10 donors were considered. Which two were excluded and why? Lines 542-544: According to this sentence, i.e., "From the sensitivity analysis, we calibrate a and b, choosing the set of 8 from 10 donors giving the lowest values of RMSE on the 15 years calibrated period", we know that 8 donors were used to determine the value of a and b. But the following sentence says you used 9 donors: "We use the 9 donors DH2, DH3, DH4, DH6, DI2, DI3, DI4, DI5 and DI6".

So it is not clear how many donors were used. Lines 548-550: Is there any specific reason for plotting the points of Etoa in Figure 11? Lines 550-552: When DH3 and DI2 are used for the first level of validation, both E are larger than 0, is that because using the calibrated a and b would overestimate the hydrological index? Lines 582-583: T2 and T8 also have different precipitation according to Figure 6d. Therefore, how much difference in runoff between T2 and T8 is caused by the difference in precipitation?

---

## Author Comment (AC1) · 28 Jul 2019

**Responses to Referee #1**

Dear Referee,

Thank you for your comments. We partially agree with referee #1 concerning the comments on the hypotheses, concepts, science questions and advancements in this paper. The scope of this paper is to develop a combined approach of data acquisition and the development of a new semi-distributed model taking into account land-use changes to reconstruct and predict annual runoff on a catchment exposed to high urban increase. As stated by referee #2, "the research question is interesting, and the methods used in this study are well explained." However, we totally agree with all the other comments of referee #1 especially that some hypotheses and concepts of the model worth detailed analysis and comparison to classical well-known approaches in the literature. We agree with the main comments of both referees #1 and #2, and we propose to substantially modify the paper outline and add new sections taking into account all points raised by both referees.

We give in this letter our responses in blue.

For the authors Camille Jourdan

**Anonymous Referee #1**

Received and published: 22 June 2019

"The paper develops a simple regression-based annual runoff model and evaluates the model's performance in data sparse environments in Cameroon to assess the role of urbanization on runoff. The model uses and catchment index I, similar to the CN of the SCS-CN equation. The difference was that this index was designed for annual runoff.

I was not able to find any new hypotheses, concepts, science questions and advancements in this paper."

We agree with referee #1 that the paper doesn't put forward well the advancements and we would like to perform significant modifications in order to improve it. We want to emphasize that the main scientific question treated by the paper is about hydrological impacts of urbanisation in data-sparse region with the interest to present new data acquisition in tropical urbanized area. This paper also proposes a new parsimonious model leading to a satisfying back-casting of annual runoff at the catchment scale.

The first interest of our study is the data collected on the field for heterogeneous land-use basins from short-term instrumentation. Many papers assessed the impact of forest conversion into cropland on the hydrological cycle in tropical countries (Beck et al., 2013; Giertz et al., 2006; Yira et al., 2016), but very few studies treats the urbanization impacts in tropical zone from field observations due to the lack of data. The data presented in the paper at the annual scale are also available at hourly scale and enable to assess the impact of land-use changes on hydrology at several time scales. We propose to make additional analyses at the monthly and seasonal scales in order to improve this paper (see response to comments  $n^3$  and  $n^4$  of referee #2). The analysis at the finer hourly time step is treated in another paper in preparation.

The second interest is the use of historical sparse data to complete the instrumentation in order to have a maximum of information to validate the model. Most of the studies in tropical regions applied simple models to assess hydrological impacts of urbanization (Barron et al., 2013; Remondi et al., 2016) but few have the opportunities to confront simulations with real field observations.

The third interest is the model itself, using sparse data, easy-to-use by stakeholders, parsimonious and integrating land-use changes. The hydrological index I is a simple indicator taking into account land-use change using available remote sensing data.

"I'd would have appreciated if the authors at least used the Budyko framework or simple annual water models like the ABC model, both of which has more sound conceptual basis that involve water and energy."

We agree that this new model must be compared to classical approaches such as the Budyko or simple conceptual models in order to show the advantages and the limitations of the new model we propose. We propose to add a special section to do this comparison: 1. The Budyko model; 2. The conceptual annual model GR1A (Mouelhi, 2003; Mouelhi et al., 2006); 3. The monthly GR2M model (Mouelhi, 2003). These models have similar conceptual concepts as the ABC model.

**1. The Budyko model**

The Budyko model (Budyko, 1974) was applied at the annual data on: i) the available data at the Mefou basin at outlet at Nsimalen (catchment area 421 km2) and ii) for the donor catchments. We included in the analysis uncertainties on precipitation, potential evapotranspiration *PET*, and losses due to overbank flow at the outlet.

Figure R1.1 shows the results at the Méfou main outlet at Nsimalen. The green dark points correspond to years before 1980 with a low impact of urbanization. The light green points correspond to years after 2000 with a high impact of urbanization. We observe that for the period before 1970, points are very close to the energy limit; the estimated actual evapotranspiration *AET* is close to the estimated potential evapotranspiration *PET*. Then, points trends to go down over time. The value of *AET/P* (with *P* the annual precipitation) is about 0.75 for 1964-1967 and below 0.6 for 2005-2013. This first analysis confirms the impact of urbanization on the annual water balance at the whole catchment scale at Nsimalen.

**Figure R1.1.** The Budyko model showing the relationship between the aridity index *PET/P* and *AET/P* for the available yearly data of the Mefou basin at outlet at Nsimalen (*PET*: annual potential evapotranspiration; *P*: annual precipitation; *AET*: actual evapotranspiration; with the hypothesis of a nil annual storage variation as discussed later in response to comment n°3 of referee #2). Green dark points correspond to the period before 1980 (low impact of urbanization) and light green to the period after 2000 (high impact of urbanization).

Figure R1.2 shows the results for the donor catchments. The dark blue points correspond to donors with low index I (low urbanization) and light blue to donors with high I (high urbanization). We

observe that points corresponding to catchments with low index I (i.e. land-use to natural conditions without urbanization) are close to the energy limit with AET/P around 0.75. More the index increases, more points trend to go down and move away from the energy limit with AET/P around 0.2. This relationship shows the high impact of the I index (which increases with urbanization) on the annual water balance.

**Figure R1.2.** The Budyko model showing the relationship between the aridity index *PET/P* and *AET/P* for the donor catchments. Dark blue points correspond to donors with low *I* index (low urbanization) and light blue to donors with high *I* index (high urbanization).

We propose to add these results on a new section (see the suggested outline at the end of this response).

**2. The annual GR1A model**

We used the *GR1A* model to compare the model we developed to a classical standard annual hydrological model used in France (Figure 12). The *GR1A* is based on the Turc (1954) equations framework (Mouelhi, 2003). GR1A is defined by the equation R1.1.

$$R_{k} = P_{k} \left\{ 1 - \frac{1}{\left[ 1 + \left( \frac{0.7P_{k} + 0.3P_{k-1}}{X.PET_{k}} \right)^{2} \right]^{0.5}} \right\}$$
(R1.1)

with k the studied year, R the annual runoff, P the annual precipitation and *PET* the potential evaporation. The GR1A model has only one parameter X which is empirical and not easily linked to urbanization. Consequently, it is difficult to apply the model in changing environment conditions.

We carried on several calibration/validation tries of the model *GR1A* on the available database of the Mefou basin at outlet at Nsimalen (Table R1.1): i) calibration of X on the period 1964-1976 before urbanization and validation on the period 2005-2013 after urbanization; ii) similar as in i) but exchanging the periods of calibration and validation; iii) calibrating on odd years and validating on even years in order to obtain a mean value of X on periods impacted by urbanization. Figure R1.3 shows the different simulations.

| Calibration
Period           | Calibrated
X | Performance
(RMSE in
mm)
Calibration
period | Validation
Period             | Performance
(RMSE in
mm)
Validation
period | Total
performance
(RMSE) |
|---------------------------------|-----------------|---------------------------------------------------------|----------------------------------|--------------------------------------------------------|--------------------------------|
| 1964 – 1976                     | 1.15            | 91.0                                                    | 2005 - 2013                      | 170.1                                                  | 144.0                          |
| 2005 - 2013                     | 0.90            | 152.7                                                   | 1964 - 1976                      | 215.0                                                  | 146.0                          |
| Odd years
over 1964-
2013 | 1.05            | 123.4                                                   | Even years
over 1964-
2013 | 130.4                                                  | 126.8                          |

Tableau R1.1. Summary of calibration tries for GR1A model at the Méfou outlet at Nsimalen.

---

## Author Comment (AC2) · 28 Jul 2019

Please find the Response to Referee #2 in the attached pdf-file

Please also note the supplement to this comment:
https://www.hydrol-earth-syst-sci-discuss.net/hess-2019-116/hess-2019-116-AC2-supplement.pdf
* * *

---

## Author Comment (AC3) · 28 Jul 2019

**Responses to Referee #2**

Dear Referee,

We are very grateful to the referee for constructive comments of the manuscript. We totally agree with all her/his recommendations and we give in this letter our responses in blue.

For the authors,

Camille Jourdan

**Anonymous Referee #2**

The scope of this paper is to develop a combined approach of data acquisition and the development of a new semi-distributed model taking into account land-use changes to reconstruct and predict annual runoff on a catchment exposed to high urban increase. The research question is interesting, and the methods used in this study are well explained.

Thank you.

However, I have some major comments listed below.

1. Lines 122-123: How about potential evaporation? The effect of potential evaporation on annual runoff may need to be incorporated into the model.

In equatorial climate, inter-annual variability of potential evaporation is quite low and variability through the year is also limited by low variation of temperature and available water. We use the global *ET* product *GLEAM* (Global Land Evaporation Amsterdam Model; Miralles et al., 2011) to estimate the potential *PET* and actual evapotranspiration *AET* since 1997 over the region of Yaoundé. This product is known to perform well is African equatorial climate (Trambauer et al., 2014). Figure R2.1a shows the mean daily *PET* and *AET* with inter-annual variability and Figure R2.1b shows the annual *PET* and *AET* estimated over the period 1997 – 2017 (estimation from *GLEAM* product). The variation of annual values of estimated *PET* is low with a minimum of about 1010 mm and a maximum of 1125 mm. In equatorial regions covered by natural humid forest, *AET* is usually very close to the *PET* because of high water availability, with for example *AET* near from 1300 mm for Congo forest where *PET* is about 1400 mm (Rodier, 1964). The simple Thornthwaite method gives an annual *PET* of about 1200 mm on the Yaounde region. In natural condition, Olivry (1979) estimated a value of *AET* of 1200 mm in the Nyong basin using the water balance method with an *AET* very close to *PET*. In natural conditions, *PET* is satisfied during the both wet seasons and the first (the smallest) dry season.

The main conclusion is low inter-annual and intra-annual variabilities of *PET* over the basin. We propose to include a deeper analysis of *PET* data in the Section 3.2 of the new suggested outline paper (see the new suggested outline at the end of this response).

[Figure]

[Figure]

a)
b)

**Figure R2.1.** a) Annual *PET* and *AET* estimated from the satellite-based product *GLEAM* over the period 1997–2017. b) Mean daily *PET* and *AET* estimated over the period 1999-2017 from the product GLEAM.

2. Line 139 equation (1): how is this equation obtained from SCS curve number method applied to the annual scale?

The proposed model (Eq.1 in presented paper) is an approximation of the annual model proposed by Ponce and Shetty (1995) on similar mathematical equations as for the SCS model.

Ponce and Shetty (1995) proposed a relationship between the surface runoff S and precipitation P such as :

$$S = \frac{(P - \lambda_s W_p)}{P + (1 - 2\lambda_s)W_p} \quad \text{if} \quad P > \lambda_s W_p \quad \text{otherwise } S = 0 \tag{R2.1}$$

with *S*, annual surface runoff; *P,* annual precipitation; $\lambda_s$ and $W_p$ two parameters.

The model has two parameters, $\lambda_s$ and $W_p$, to be calibrated using annual data. The model suggests that if precipitation is lower than a threshold ($\lambda_s W_p$) then S = 0, while if precipitation is superior than the threshold Eq.(R2.1) is used.

After separation of baseflow and surface runoff from total runoff by two parameters Boughton method (Boughton, 1993; Chapman, 1999), we calibrated the two parameters of the Eq.(R2.1) for the period 1965-1978 for the Mefou basin at Nsimalen outlet (421 km²). We obtain $\lambda_s = 0.05$ and $W_p = 9830$ mm. Figure R2.2 presents the calibrated Eq.(R2.1) on the observed annual runoff for the period 1965-1978 on the Mefou basin at Nsimalen outlet.

[Figure]

**Figure R2.2.** Simulation of surface runoff *S* for the Méfou basin at Nsimalen outlet, using Ponce and Shetty (1995) model (Eq. R2.1 noted PS_S on the Figure) calibrated on observed annual surface runoff (red points) for the period 1965-1978.

For the baseflow $U$, Ponce and Shetty (1995) proposed the following relationship:

$$U = \frac{(P - \lambda_u V_p)}{P + (1 - 2\lambda_u)V_p} \quad \text{if} \quad P > \lambda_u V_p \quad \text{otherwise } U = 0 \tag{R2.2}$$

with $\lambda_u$ and $V_P$ the two parameters.

The total annual runoff simulated is :

$$R = U + S \tag{R2.3}$$

We calibrated the two parameters of the Eq.(R2.3) for the period 1965-1978 for Mefou basin at Nsimalen outlet (421 km²); we obtain $\lambda_u = 0.01$ and $V_p = 5300$ mm.

[Figure]

**Figure R2.3.** Simulation of baseflow $U$ for the Méfou basin at Nsimalen outlet using Ponce and Shetty (1995) model (Eq. R2.2 noted PS_U on the Figure) calibrated on observed annual surface runoff (red points) for the period 1965-1978.

In our study case, annual precipitation ranges between a minimum ($P_n$) and a maximum ($P_x$) values, and surface runoff is observed for all the years. We extrapolate the relationship to the total runoff by simplifying the SCS Eq(R2.1) to a simple polynomial relationship $AP^2 + BP$ (Eq.1 of the proposed paper).

Figure R2.4 shows the model estimation of total annual runoff for the Mefou basin at Nsimalen outlet for the period 1965-1978(few impacted by land-use changes) by :

-   The four parameters model ($\lambda_s$, $W_p$, $\lambda_u$ and $V_P$) of Ponce and Shetty (1995), noted PS_U+S (EqR2.1 to R2.3)

-   The simplified two parameters model from Ponce and Shetty (1995) equation, noted PS*, extrapolating the Eq.(R2.1) from an application to surface runoff $S$ to the total runoff $R$.

-   The proposed two parameters model (Eq.1 of the proposed paper), only operable between $P_n$ and $P_x$ but leading to very similar results to the two models PS_U+S and PS*.

[Figure]

**Figure R2.4.** Simulation of the total runoff R using: i) the four parameter Ponce and Shetty (1995) model (PS_U+S); ii) the two parameters modified Ponce and Shetty (1995) mode (PS*); iii) the model proposed in Eq.1 of the proposed paper. All three models are calibrated on the observed annual runoff for the Méfou basin at Nsimalen outlet.

We propose to add a comparison of the proposed model (Eq.1) with different formulations of annual rainfall-runoff models (PS_U+S, PS*) in the section 4.2 of the new suggested outline paper in order to show the validity of the proposed equation for the study site.

3. The effect of inter-annual soil water storage carry-over is not considered. As shown in Lines 357-358, there are humid (1960-1970, 1980-1990, and 2006-2013) and dry (1935-1950, 1970-1980, and 1990-2000) periods. The storage carry-over from the transition between wet and dry periods may be not negligible.

We agree that the storage variation between humid and dry years worths further analysis.

First we studied the precipitation at a daily time step. Figure R2.5 shows an analysis of the precipitation seasonality over the period 1950–2014 comparing the 11-day moving average value of the 10-wettest and the 10-driest years. Note that there are two wet seasons (in blue) and two dry seasons (in orange). We observe that the second dry season is the longest and the driest of both dry seasons. Therefore, we decided to set the beginning of the hydrological year at the 1st of March, directly after the main dry season. The inter-annual variability of precipitation is quite low for the second dry season as shown in Figure R2.5 presenting the seasonality for the ten wettest and the ten driest years. The low precipitation variability on the main dry season led us to make the hypothesis that inter-annual soil water storage variation is small and doesn't play a key role in annual runoff value.

[Figure]

**Figure R2.5.** Seasonal precipitation for the period 1950-2014:11-day sliding average value of the ten wettest years and the ten driest years. Note that there are two wet seasons (in blue) and two dry seasons (in orange).

Second, and in order to test the hypothesis of negligible inter-annual soil water storage variation, we used the monthly model *GR2M* (Mouelhi, 2003). The GR2M model structure is available at following url link: https://webgr.irstea.fr/modeles/mensuel-gr2m/

The model structure is shown below:

[Figure]

The model has two reservoirs, the reservoir S for production and the reservoir R for routing. The soil storage is represented by the depth S of the reservoir S. The model has as input monthly precipitation and potential evapotranspiration. The model simulates monthly discharge and has two parameters to be calibrated: $X_1$ which represents the depth of the reservoir S and $X_2$ which represents the flow exchanged outside the catchment. GR2M was largely applied on west and central Africa (Louvet et al., 2016; Ardoin-Bardin et al,. 2005).

The model was first calibrated on the period 1968-1978 with low impact of urbanization, and we obtained a Nash-Sutcliffe Efficiency criteria $NSE = 0.72$, with $X_1 = 8.50$ and $X_2 = 0.67$ (Figure R2.6). Figure R2.7 shows the variation of the inter-annual storage component (Delta Storage $\Delta S$ = difference of the levels of reservoir S in Figure R2.8 between the end and the beginning of the hydrological year at the end of February). Figure R2.8 shows the monthly variation of the soil storage component for the calibrated period. For these ten studied years, the average annual variation $\Delta S$ is about 59 mm with a mean annual precipitation $P$ of 1600 mm; this result $\Delta S / P < 3$ % confirms the low impact of inter-annual soil water storage variation. The second dry season has very low precipitation, and most of the annual storage is consumed by evaporation and discharge. However some specific years could present higher inter-annual soil water variation that could impact the annual runoff.

We observe a low inter-annual variation of soil water storage leading to the hypothesis to neglect this term in the first version of proposed model. In a revised version, we will make a sensitivity analysis of the model taking into account $\Delta S$ which can be added/subtracted to annual rainfall.

[Figure]

a)                                                                    b)

**Figure R2.6**. The monthly hydrological model GR2M applied on the Méfou at Nsimalen. a) Monthly precipitation, and the observed and simulated monthly discharges. b) Comparison of simulated and observed monthly discharge. Qsim and Qobs are respectively the simulated and the observed annual runoff for the Mefou basin at Nsimalen outlet.

[Figure]

**Figure R2.7.** Inter-annual values of the variation of the storage components ($\Delta S$) of the GR2M model on the period 1968-1978 using the calibrated parameters (with $NSE = 0.72$).

[Figure]

**Figure R2.8.** Monthly values of storage components (*S*) of the GR2M model over the period 1968-1978 from calibrated parametrization over monthly discharge (*NSE* = 0.72). We observe the low variation of the soil water storage at the end of the hydrological year by the end of February.

We propose to add a deeper analysis of available data integrating the inter-annual variation of soil water storage in the Section 3.4 and 3.5 of the new proposed outline.

The GR models (GR1A and GR2M) are not adapted to take into account land-use changes over time. As presented in the response to comment n°3 of referee #1, calibrated parameter X of the GR1A model varies from 0.90 for the period 1964-1976 to 1.15 for the period 2005-2013. Similarly, the calibrated parameters of GR2M ($X_1$ and $X_2$) changes with urbanization and land-use changes. That's why there is a need to a simple parsimonious model taking into account land-use change as suggested in the model developed.

4. Lines 358-362: The change of seasonal rainfall pattern will affect the annual runoff. It will be good to explore or discuss the potential impact.

Changes in seasonal rainfall are highlighted in Figure R2.9a with the analysis of two seasonal patterns for the period 1964–1976 with no impact of urbanization, and a recent period 2005 – 2014 with high impact of urbanization. For both periods, we present the 11-day sliding average (five days before and five days after) of the daily average precipitation. The analysis shows that the first dry season (noted DS-I) trends to be shortened with more precipitation during August for the recent years. A smaller change can be observed on the second dry season (noted DS-II) with less precipitation for this already very dry season. These limited changes observed in seasonal precipitations pattern don't enable to explain very high changes in seasonal discharge observed between the two periods as shown in Figure R2.9b. Impacts of observed land-use changes over the discharge seems widely override changes in seasonal precipitation. Even if impacts of changes in precipitation seem limited compared to the changes in land-use, both changes can be combined for the first dry season with a reduction of the length of the dry season before the main wet season (WS-II). One of the impacts can be a higher saturation of soil and groundwater storage before the main humid season with impacts on high flows, overflows and the total volume of runoff.

[Figure]

[Figure]

a)                             b)

**Figure R2.9.** a) Seasonal precipitation for 1964-1976 and for 2005-2014 at the historical raingauge P1 (See the location on Figure 3). b) Seasonal runoff at the Mefou basin outlet at Nsimalen for 1964-1976 and 2005-2013. We observe two wet seasons in blue (WS-I and WS-II) and two dry season in orange (DS-I and DS-II).

Changes in seasonal precipitation seem quite limited and don't let to explain major changes observed for discharge. However, precipitation changes could be combined with land-use changes and could participate to increase impact on discharge because of land-use changes. We propose to add a deeper analysis of available data integrating seasonal rainfall changes impacts in the section 3.5 of the new proposed outline (see the outline proposed at the end of this letter).

5. Figure 7: Considering the annual runoff coefficient, the uncertainty for the linear model between runoff coefficient and precipitation is large (i.e., the scatter of data points around the line). This is due to the following potential reasons:

Uncertainties observed on Figure 7 should also be due to uncertainties on input data of precipitation and discharge. In this sparse-data context, the use of only two available rain-gauges for a catchment of 421 km² (for the Mefou basin outlet at Nsimalen) may lead to significant uncertainties because of the (limited) heterogeneity of spatial precipitation and its inter-annual variabilities. Data presented in Figure 7 are the only available data letting us to assess the value of $\beta_1$. Despite of the uncertainties due to precipitation estimation, we notice clear increasing trends of the runoff coefficient $\rho$ with the precipitation $P$, and the regression enables to calculate $\beta_1 = 15\%$ at Nsimalen (red points). Increasing trends is also observed for Etoa (blue points) and Mfoundi upst (green points). Due to uncertainties of precipitation on Nsimalen and most of the points of Etoa (except for one year), we didn't use these data in donors database but only in validation (Figure 11 for Etoa, Figure 12 for Nsimalen).

1) the validity of equation (1);

See details in the response to comment n°2.

2) the effect of potential evaporation;

As answered for the comment n°1, *PET* is characterised by very low inter-annual variability of about 100 mm but could explain part of the uncertainty observed in Figure 7. *PET* is difficult to estimate and varies a lot depending of the global product used. An annual value of a constant *PET* = 1200 mm seems to be a first satisfying assessment.

3) the effect of inter-annual soil water storage change;

As answered for the comment n°2, inter-annual soil water storage seems limited because of the long dry season before the beginning of the hydrological year. Moreover, the variability of precipitation over the second dry season is very small and limits inter-annual variability in soil water storage. In order to assess the impact of potential higher inter-annual soil water storage, we analysed runoff coefficient $\rho$

of years with comparable yearly precipitation values $P$ but presenting contrasting antecedent precipitations values (noted $Pa$ and characterized by the annual precipitation of the preceding year) in homogeneous land-use conditions.

Years 1967 and 1969 are characterized by comparable annual precipitation values with $P_{67} = 1622$ mm and $P_{69} = 1581$ mm but by very contrasting antecedent precipitation with respectively $Pa_{67} = 2180$ mm and $Pa_{69} = 1715$ mm. Both years present very comparable $\rho$ values with respectively 0.28 for 1967 (with more humid antecedent conditions) and 0.32 for 1969 (with drier antecedent conditions).

 4) the effect of seasonal variation of precipitation.

The seasonal variability of precipitation is presented in Figure R2.5 with seasonal precipitation for the ten wettest years, the ten driest years and the whole period. Variability is lower for the second humid season than for other seasons limiting the impact of inter-annual soil water storage. Higher variability is observed for the second dry season and may have an impact on rainfall-runoff response on the second humid season.

In order to assess the impact of variability on seasonal precipitation over "rainfall-runoff" response, we select, over the available period 1964-1976, years with annual precipitation between 1550 mm and 1650 mm. Eight years were selected which present high heterogeneities of seasonal precipitation variability. For example, the year 1973 presents a high portion of precipitation during the first humid period (52%) compared to the year 1968 (38%) but they present the same annual runoff coefficient (0.27). The year 1967 present very few portions of precipitation during both dry seasons (7%) compared to the year 1973 (21%).

However, runoff coefficients varied only from 0.25 to 0.33 without a significant impact of seasonal variability. However, the highest value of $\rho$ (0.33 for 1970) corresponds to a high portion of precipitation over the two humid seasons (87% against 83% in average) and could explain part of the uncertainty observed in Figure 7.

In a revised version, we propose to compare various versions of the model (and/or the index I) taking into account all points suggested by referee #2: model equation, PET, $\Delta$S and seasonality (in the sections 5 and 6 of the new proposed outline at the end of this letter).

6. In this paper, the temporal change of I is used to model the effect of land use change. However, agree the land use change may be better to be reflected in the parameters of the rainfall-runoff relation (e.g., A and B parameters). What are the estimated values of A and B? The parameters A and B have physical meanings. For example, soil water storage capacity is an important parameter in the SCS curve number method, and it is a function of land cover and land use. Land use change will cause the change of soil water storage capacity.

We agree that this point is a hypothesis of the model in order to simplify the model. The index $I$ is linearly linked to the parameter $B$ (See Eq. 19 of the paper), but $A$ is not impacted by the variation of $I$. The index $I$ depends only on the value of the parameter $\beta_1$ and the extreme precipitation values $P_x$ and $P_n$ (Eq.19). The parameter $A$ is the slope of the relation between precipitation and runoff coefficient, its unit is the $mm^{-1}$, and $B$ is dimensionless. For the study case of the paper, $A$ is equal to $1.3 \times 10^{-4}$ $mm^{-1}$ and $B$ is characterize by the following equation, with I varying between 0 and 1.

$$B = 0.74 \cdot I - 0.09$$

B is ranging from -0.09 and 0.65.

In a revised version, we propose to compare the present version of the model (which is simple and parsimonious) to various versions of the model structure: i) A depending on I and B independent of

I; ii) A and B depending on I. This analysis will be incorporated in the sections 5 and 6 of the new proposed outline at the end of this letter.

**Some minor comments:**

Lines 29 and 117: kmˆ2? Same for other places.

Ok.

Line 30: change "an" to "a"

Ok.

The Abstract section could be shortened.

Ok, the Abstract will be shortened.

Line 59: change "anthropic" to "anthropogenic"?

Ok.

Line 72: delete "people"

Ok.

Line 95: change "as" to "of"?

Ok.

Line 103: change "recently" to "recent"

Ok.

Line 124: "short-terms" to "short-term"?

Ok.

Line 139: APˆ2

Ok.

Line 175 and Figure 1: Why is $\beta\_2$ similar for all values of P?

As a first hypothesis, we supposed that the impact of the index *I* on runoff coefficient is independent from the precipitation *P*. The donors' database doesn't present enough element to study the impact for different ranges of precipitation.

Why are the upper bound and lower bound of Figure 1a in parallel?

Because β_2 is supposed similar for all values of *P*. In a revised version, we propose to compare various hypotheses of the model (see details in the response to comment n°6).

Line 396: Px=200mm? May be a typo.

Yes, it's a typo. *Px* = 2200 mm.

Line 433: change "(U" into "U"

Ok.

Line 461: "quite clear in term of runoff with for D14 runoff value up to 160 mm. . ." There is a typo, it should be D12 not D14.

Ok.

Line 499: A " " is missing after "consequently".

Yes, a "ρ" is missing.

Line 531: There are 12 donors totally, but only 10 donors were considered. Which two were excluded and why?

The donor $D_{I1}$ measures discharge from the dam for the hydrological year 2017-2018, and is only used as a reference to estimate the discharge value from the area controlled by the dam after its construction in 1970. The donor $D_{I6}$ covers only the period from September to December 2017 and should not be included in the donors. This donor cannot be integrated in the model but enables to analyse the hydrological behaviour of catchments with low slope by comparison with similar donors with other characteristics ($D_{I2}$).

Lines 542-544: According to this sentence, i.e., "From the sensitivity analysis, we calibrate a and b, choosing the set of 8 from 10 donors giving the lowest values of RMSE on the 15 years calibrated period", we know that 8 donors were used to determine the value of a and b. But the following sentence says you used 9 donors: "We use the 9 donors DH2, DH3, DH4, DH6, DI2, DI3, DI4, DI5 and DI6". So it is not clear how many donors were used.

We agree that this sentence is not clear, there are some typos.

The sensitivity analysis shows that using only 8 of the 10 donors leads to satisfying results. We decided to select 8 donors and we used the remaining two donors for the first validation step. So, this is an error in the text, we selected 8 donors and not 9 donors.

We selected randomly these following 8 donors: $D_{H1}, D_{H2}, D_{H4}, D_{H5}, D_{H6}, D_{I3}, D_{I4}$ and $D_{I5}$

We used the following two donors in validation: $D_{H3}, D_{I2},$

Lines 548-550: Is there any specific reason for plotting the points of Etoa in Figure 11?

The points plotted of Etoa were not all used as donors due to high uncertainties in precipitation. However, we noticed that the points plotted for Etoa in Figure 11 are very close from the estimated *I* value at Etoa (0.11) for all the points (0.08 to 0.13).

Lines 550-552: When DH3 and DI2 are used for the first level of validation, both E are larger than 0, is that because using the calibrated a and b would overestimate the hydrological index?

The donors $D_{I2}$ and $D_{H3}$ are both located below the regression line on Figure 9 leading to an overestimation of annual runoff when they are used in validation. If the hydrological index is overestimated, it is not caused by the parameters *a* and *b*, but by the procedure of hydrological index construction.

We propose to select one donor below and another one above the regression line in the new version.

Lines 582-583: T2 and T8 also have different precipitation according to Figure 6d. Therefore, how much difference in runoff between T2 and T8 is caused by the difference in precipitation?

If we apply for both target basins a precipitation of 1600 mm (mean annual precipitation over the Mefou basin), the annual runoff difference between T2 and T8 (with land-use condition of 2017) is about 265 mm. After an adjustment of precipitation according to heterogeneities observed for some specific years the annual runoff difference increase to 450 mm. Heterogeneities of precipitation and land-use are combined and lead to major differences of annual runoff.

**In summary**

We propose to substantially modify the paper outline taking into account the points stated by both referees. We suggest the new following outline

1. Introduction
2. The study site
3. What we learn from data
   3.1 Precipitation
   3.2 PET temporal analysis
   3.3 The Budyko model
   3.4 The annual *GR1A* and the monthly *GR2M* models
   3.5 Constraints for modelling
4. The annual rainfall-runoff model
   4.1 General model structure
   4.2 Sensitivity analysis and comparison of different formulations
5. The hydrological index
   5.1. The present model
   5.2. Comparison of different approaches taking into account AET, ΔS, and seasonality.
6. Applications
   6.1. The present model
   6.2. Comparison of different approaches of the model structure and the index I

In comparison to the original version, we propose to add sections and to reverse the order of other sections:

First, the introduction will be substantially reinforced in order to show the main novelty and sciences questions of this work.

Second, the study site is presented (similar to the original paper)

Third, we propose to add a detailed data analysis section called "3. What we learn from data". In this section, we keep the subsection on the spatio-temporal analysis of precipitation (noted 3.1). We propose to add four new sub sections: 3.2 PET temporal analysis (see response to referee #2); 3.3. The Budyko model (see response to referee #1); 3.3. The annual GR1A model (see response to referee #1)

and the GR2M model (see response to referee #2); 3.4. Constraints for modelling explaining the originalities of the new approach in comparison to classical ones (see responses to both referees #1 and #2).

Fourth, we propose to modify substantially the section concerning the model in order to show the hypotheses of the new approach (see response to referee #2) and to add a sensitivity analysis and a comparison between the different modelling approaches (see responses to referees #1 and #2).

In Sections 5 and 6, we propose to compare various structures of the model taking into account the model general equations, but also introducing potential evapotranspiration PET, inter-annual stock variation and seasonality (see details in the responses to referee #2).

The conclusion will be modified and adapted.

**References**

Ardoin-Bardin S., Dezetter A., Servat E., Mahé G., Paturel J.E., Dieulin C., Casenave L., 2005. Évaluation des impacts du changement climatique sur les ressources en eau d'Afrique de l'Ouest et Centrale. AISH Pub. 296, 194-202.

Boughton, W.C., 1993. A Hydrograph-based model for estimating the water yield of ungauged catchments. Hydrology and Water Resources Symposium, Newcastle, IEAust.

Chapman, T., 1999. A comparison of algorithms for stream flow recession and baseflow separation. Hydrological Processes 13, 701–714. https://doi.org/10.1002/(SICI)1099-1085(19990415)13:5<701::AID-HYP774>3.0.CO;2-2

Louvet, S., Paturel, J.E., Mahé, G., Rouché, N., Koité, M., 2016. Comparison of the spatiotemporal variability of rainfall from four different interpolation methods and impact on the result of GR2M hydrological modeling—case of Bani River in Mali, West Africa. Theor Appl Climatol 123, 303–319. https://doi.org/10.1007/s00704-014-1357-y

Miralles, D.G., Holmes, T.R.H., De Jeu, R.A.M., Gash, J.H., Meesters, A.G.C.A., Dolman, A.J., 2011. Global land-surface evaporation estimated from satellite-based observations. Hydrology and Earth System Sciences 15, 453–469. https://doi.org/10.5194/hess-15-453-2011

Mouelhi, S., 2003. Vers une chaîne cohérente de modèles pluie-débit conceptuels globaux aux pas de temps pluriannuel, annuel, mensuel et journalier (thesis). Paris, ENGREF.

Olivry, J.-C., 1979. Monographie du Nyong et des fleuves cotiers : T.I.: Facteurs conditionnels des régimes hydrologiques. T.2: Hydrologie du Nyong. T.3: Hydrologie des fleuves cotiers. ONAREST, Yaoundé.

Rodier, J., 1964. Régimes hydrologiques de l'Afrique Noire à l'Ouest du Congo, Mémoires ORSTOM. ORSTOM, Paris.

Trambauer, P., Dutra, E., Maskey, S., Werner, M., Pappenberger, F., van Beek, L.P.H., Uhlenbrook, S., 2014. Comparison of different evaporation estimates over the African continent. Hydrology and Earth System Sciences 18, 193–212. https://doi.org/10.5194/hess-18-193-2014